# Communication Efficient Federated Learning via Model-Agnostic Projection Adaptation

## Abstract

Federated learning (FL) enables collaborative model training across distributed clients without centralizing sensitive data. Despite recent advancements, communication overhead remains a major bottleneck, particularly for large-scale models. Utilizing low-rank adaptation (LoRA) techniques can mitigate this challenge by decomposing each layer into a *reconstruction matrix* and a *projection matrix*, and transmitting either both matrices or only the projection matrix while keeping the reconstruction matrix fixed. While effective, these techniques operate on individual layers, are architecture-dependent, and suffer from performance limitations due to their fixed reconstruction matrix. We propose Model-Agnostic Projection Adaptation (MAPA), a novel factorization approach that treats the entire model parameter space as a single matrix rather than decomposing layers independently. MAPA introduces round-wise randomization of the reconstruction matrix to avoid suboptimal solutions while flexibly balancing communication and accuracy. MAPA also reduces the memory and computational overhead relative to LoRA, ensuring efficiency in both communication and computation when applied to federated learning. Empirical results demonstrate the effectiveness of MAPA in various FL settings.

## 1. Introduction

Federated learning (FL) is a distributed framework that enables training across many devices (clients) without centralizing data. In a typical FL process, each client downloads a model from the server, trains it using local data, and then uploads the updated model back to the server. The server constructs the global model through the aggregation of lo-cal updates, e.g., federated averaging (FedAvg) (McMahan et al., 2017). This iterative procedure is carried out through several rounds of communication, allowing clients to collaboratively enhance the model. While FL offers many advantages, a significant obstacle is the high communication overhead from exchanging model updates between clients and the server. This problem is especially noticeable with resource-constrained clients and large-scale models.

To address the communication burden in FL, various strategies have been developed to reduce either the communication frequency (Stich, 2018; Sattler et al., 2019; Li et al., 2020) or the communication load per round (Konečný, 2016). Methods aimed at reducing the communication load per round are broadly categorized into *sketched updates*, which involve optimizing the local model followed by gradient compression, and *structured updates*, which directly train in a lower-dimensional subspace such as random masks, weight-sharing, and low-rank factorization.

Low-rank adaptation (LoRA) has demonstrated remarkable success in adapting large-scale pre-trained models by reducing the number of parameters through low-rank factorization (Hu et al., 2021; Ou et al., 2023; Bertsimas et al., 2023). Specifically, LoRA approximates each layer by using a *reconstruction matrix* $A$ and a *projection matrix* $B$ given a fixed rank. Recently, researchers have extended the idea of LoRA to FL settings to handle the communication burden issue (Yi et al., 2023; Sun et al., 2024; Cho et al., 2024; Kuo et al., 2024; Yang et al., 2024; Qi et al., 2024). However, choosing an excessively low rank can lead to performance degradation, while choosing a large rank does not provide significant gains in communication efficiency. More recent works on *Freeze A LoRA* (FA-LoRA) address this issue by freezing the reconstruction matrix $A$ and communicating only the projection matrix $B$, further reducing communication overhead (Sun et al., 2024; Zhang et al., 2023; Zhu et al., 2024; Hao et al., 2024). The first two subfigures in Figure 1 compare the ideas of LoRA and FA-LoRA.

**Challenges.** However, while LoRA and FA-LoRA offer notable advantages, they continue to face several key challenges: The layer-wise low-rank factorization imposes architecture-specific constraints, limiting the trade-off between communication, computation, and performance.

[1]Anonymous Institution, Anonymous City, Anonymous Region, Anonymous Country. Correspondence to: Anonymous Author <anon.email@domain.com>.

Preliminary work. Under review by the International Conference on Machine Learning (ICML). Do not distribute.

Figure 1: Comparison of LoRA, FA-LoRA, and MAPA, highlighting differences in parameter updates and matrix factorization.

Moreover, in FA-LoRA, freezing $A$ restricts the model's ability to explore richer subspaces, often leading to suboptimal solutions (Guo et al., 2024). Thus, we aim to answer the following key question:

*How can we develop an adaptive and generalizable approach to address both the performance suboptimality of FA-LoRA and communication suboptimality of LoRA while overcoming the rigidity of architecture-dependent designs?*

**Key Idea.** We propose Model-Agnostic Projection Adaptation (MAPA), a new approach that reimagines low-rank adaptation in FL. The key idea is to treat the entire model update as a single matrix rather than factorizing parameters layer by layer, as depicted in Figure 1: MAPA reshapes the *universal update vector* $\Delta W \in \mathbb{R}^{d \times 1}$ to $\Delta W \in \mathbb{R}^{\lceil \frac{d}{k} \rceil \times k}$ and further decomposes it into a reconstruction matrix $A \in \mathbb{R}^{\lceil \frac{d}{k} \rceil \times 1}$ and a projection vector $B \in \mathbb{R}^{1 \times k}$, where $d$ is the total number of parameters and $k$ is a design parameter. This design choice eliminates architecture-specific constraints, making MAPA applicable to any model architecture while also reducing computational costs. Additionally, instead of being locked into a frozen reconstruction matrix $A$, MAPA explores new subspaces in every federated round by randomly generating the reconstruction matrix, reducing the risk of getting trapped in suboptimal parameter spaces.

**Summary of Contributions.** Overall, the integration of (i) single matrix update/factorization, (ii) training only the projection vector, and (iii) exploring new subspaces through repeated reconstruction matrix generation in MAPA provides a versatile mechanism to balance communication costs and performance while being computationally lighter than FA-LoRA. Figure 1 illustrates how MAPA differs from LoRA and FA-LoRA, highlighting its model-agnostic nature and parameter reduction. Our key contributions are as follows:

- We introduce **model-agnostic projection adaptation (MAPA)** that streamlines LoRA implementation, enhances computation and communication efficiency, boosts performance by exploration and offers flexibility in balancing communication and error rate.

- We analyze the convergence behavior of MAPA and also demonstrate that under certain conditions, model-agnostic factorization preserves the same error rate as layer-wise factorization while requiring less computational burden.

- We conduct extensive experiments across diverse datasets, model architectures, and baselines, demonstrating that MAPA surpasses existing methods.

## 2. Background and Related Works

In this section, we explore various ideas that have been developed to address communication efficiency in FL. These strategies generally focus on reducing either the communication frequency or the communication load per round. To decrease communication frequency, methods such as performing multiple local epochs on clients (Stich, 2018) and selecting a subset of clients to participate in each training round (Sattler et al., 2019; Li et al., 2020) have been proposed. On the other hand, methods aiming to reduce the communication load per round have been studied more extensively. These methods can be divided into two categories of *sketched updates*, and *structured updates* (Konečnỳ, 2016).

Here, we first explore the sketched update techniques, highlighting methods that project the gradient in a subspace and discussing their shortcomings to structured update techniques that train the gradient directly in the subspace. Afterward, we look into structured update techniques and focus on LoRA-based methods in communication-efficient FL to highlight the novelty and advantages of our work compared to recent studies. Although these strategies are complementary in practice and can jointly enhance the scalability and efficiency of FL, this work evaluates them individually to provide a detailed analytical comparison of each approach.

**Sketched Update.** Sketched update is a two-step method, where first, the full space gradient is computed, and then, it is projected into a subspace. It includes techniques such as sparsification (Konečnỳ, 2016), quantization (Alistarh et al., 2017; Mao et al., 2022), gradient subspace projection (Azam et al., 2021; Oh et al., 2022; Park & Choi, 2023),

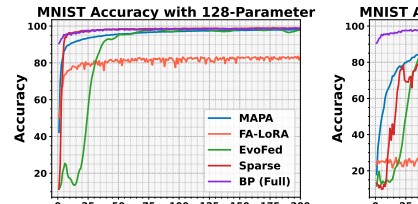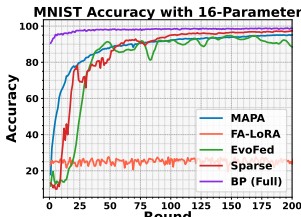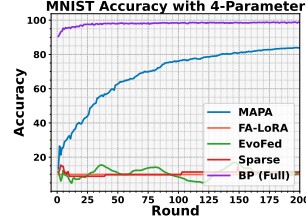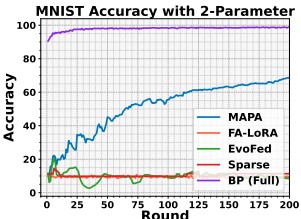

Figure 2: Performance comparison in a centralized setting with varying trainable parameters on the MNIST dataset.

and random subspace projection (Rahimi et al., 2024; Shi & Eryilmaz, 2021). The concept of subspace projection methods is that for a given gradient $\mathbf{g} \in \mathbb{R}^d$, reconstruction matrix $A \in \mathbb{R}^{d \times p}$, find the projection vector $B \in \mathbb{R}^p$, which minimizes the reconstruction error $\|\mathbf{g} - AB\|_2$, where $d$ denotes the total number of model parameters and $p \ll d$.

$$B^* = \arg \min_{B \in \mathbb{R}^p} \|\mathbf{g} - AB\|_2 \quad ; \quad B^* \approx \mathbf{A}^\top \mathbf{g}.$$

**Sketched Update Limitations.** Although sketched methods benefit from a high-quality gradient $\mathbf{g}$, one of their shortcomings is blindness to the loss surface $\mathcal{L}(W; \mathcal{D})$ and alternative solutions beside $\mathbf{g}$ that can be reconstructed more accurately from the projection subspace. They typically perform well, given a communication budget that is large enough. However, as the communication rate decreases, the reconstruction of the projection vector ends up far off from the original gradient $g$. In contrast, subspace optimization leverages the loss surface of data to find the steepest direction within the subspace, leading to a more effective reduction in loss. Figure 2 presents an example of centralized MNIST training, illustrating the performance degradation of sketched update techniques such as EvoFed (Rahimi et al., 2024) and Top-$k$ Sparsification compared to MAPA. As sparsity increases, MAPA continues to converge, even when limited to just 2 or 4 trainable parameters out of 11,274 dimensions. Similarly, FA-LoRA is shown to converge to suboptimal solutions even in centralized settings. This occurs because FA-LoRA is restricted to a fixed subspace, optimizing only within that constraint while remaining blind to the full parameter space. In contrast, MAPA mitigates this limitation by introducing randomization at each training epoch, allowing exploration beyond a static subspace.

**Structured Update.** These methods reduce the number of trainable parameters that need to be optimized and communicated by constraining the parameter space. These methods include low-rank adaptation (Cho et al., 2024; Sun et al., 2024; Kuo et al., 2024; Yi et al., 2023; Yang et al., 2024; Qi et al., 2024), pruning (Luo et al., 2017; Zhang et al., 2018), and weight-sharing (Ullrich et al., 2017). Among these, the low-rank approximation is widely used because of its solid theoretical foundation and ease of hardware implementation (Liu et al., 2022; Wang et al., 2018; Jaderberg et al., 2014; Lebedev et al., 2014; Denil et al., 2013).

**Low-Rank Adaptation (LoRA).** LoRA is a practice to approximate each layer's large-weight tensors by the product of smaller ones, consequently reducing the number of trainable parameters of each layer. Therefore, it is dependent on the layer's architecture and requires a careful network design that considers a specific factorization rank and implementation for each layer. In contrast, MAPA introduces a **global model-agnostic factorization** independent of the model architecture by viewing the gradients of all layers as a single matrix. MAPA does not require any adjustment to the network architecture, which not only simplifies the implementation but also gives more control over the size of factorization matrices while reducing the overall computation as well as the total size of the factorized matrices.

**FA-LoRA.** Recent approaches have further enhanced LoRA by freezing the reconstruction matrix $A$ and only updating the projection matrix $B$ (Sun et al., 2024; Zhang et al., 2023; Zhu et al., 2024; Hao et al., 2024). The suboptimality of these methods comes from optimizing in a fixed subspace $A$ (Guo et al., 2024). The FedSA-LoRA (Guo et al., 2024) addresses this problem with the share-$A$ LoRA methodology, which trains both $A$ and $B$ but only shares the matrix $A$. However, this solution is not efficient in terms of communication. In contrast, MAPA allows the exploration of new subspaces at each round without additional communication overhead by resetting $B = \mathbf{0}$, regenerating $A \sim \mathcal{N}(0, I)$, and updating the model parameters in every FL round.

In summary, although existing low-rank factorization methods offer a promising approach to improving communication efficiency in FL, they remain highly architecture-dependent and prone to suboptimality. To address these limitations, we propose a global model-agnostic factorization technique, which enhances flexibility, reduces computational and memory overhead, and achieves higher performance by promoting the exploration of subspaces.

## 3. Proposed Method

In this section, we present MAPA and its application in FL. We start by elaborating on the MAPA factorization technique and illustrate its key characteristics in terms of communication overhead and error rate. Subsequently, we describe the process for leveraging MAPA factorization within the FL process.

Figure 3: Step-by-Step illustration of methodology based on propositions, demonstrating how each step will contribute to designing MAPA factorization and differing from LoRA architecture.

## 3.1. Model-Agnostic Projection Adaptation (MAPA)

Recent literature studied the effectiveness of LoRA given fixed reconstruction matrix on FL communication efficiency (Sun et al., 2024; Zhang et al., 2023; Zhu et al., 2024; Hao et al., 2024). The common idea is to factorize the model update as $\Delta W_i = A_i B_i$, where $A_i \in \mathbb{R}^{d_1 \times q}$ and $B_i \in \mathbb{R}^{q \times d_2}$ for $q < \min(d_1, d_2)$ in each layer $W_i \in \mathbb{R}^{d_1 \times d_2}$. They take advantage of freezing the reconstruction matrix $A_i$, limiting the trainable parameters to the projection matrix $B_i$ for each layer $i$, thus reducing communication. However, this layer-wise FA-LoRA approach suffers from suboptimality and restricted rank due to architecture-dependent factorization and also requires a specific design for factorization matrices.

**MAPA Description.** MAPA represents a black-box factorization that is agnostic to the model. Layer architectures are not limited to one frozen subspace of optimization and can continuously explore the parameter space toward the global optimum. As illustrated in Figure 1, MAPA reshapes the *universal vector* $\Delta W \in \mathbb{R}^{d \times 1}$ to $\Delta W \in \mathbb{R}^{\lceil \frac{d}{k} \rceil \times k}$ and further decomposes into a reconstruction vector $A \in \mathbb{R}^{\lceil \frac{d}{k} \rceil \times 1}$ and a projection vector $B \in \mathbb{R}^{1 \times k}$, where $k \leq d$ and $d$ is the total number of model parameters. Figure 3 presents a step-by-step illustration of MAPA methodology, as each proposition transformation can be seen as a step toward MAPA methodology. The proofs for Definition 3.3 and Propositions 3.4 to 3.6 are located in Appendix C.

**MAPA Properties.** MAPA aims to construct an expressive subspace, enabling a small $B$ to encode sufficient information for updating the model efficiently. First, we formally define the concepts of communication overhead rate and reconstruction error rate in the context of matrix factorization in Definitions 3.2 and 3.3. Using these definitions, Proposition 3.4 establishes that reshaping a single layer preserves both the factorization error and communication rates. Extending this, Proposition 3.5 demonstrates that vectorizing multiple layers into a single matrix similarly maintains these properties. Finally, this leads to the proof of Proposition 3.6,

which introduces a computationally and communication-efficient, model-agnostic factorization method as an alternative to traditional FA-LoRA techniques.

**Assumption 3.1** (**Gaussian Matrices are Full Rank**). Let $A \in \mathbb{R}^{m \times n}$ be a random matrix with entries drawn independently from a Gaussian distribution $\mathcal{N}(0, \sigma^2)$. Then, $A$ is almost surely of full rank, i.e., $\text{rank}(A) = \min(m, n)$, as the probability of $A$ being rank deficient is zero. This result follows from standard properties of random matrices (Vershynin, 2018; Tao, 2012).

**Definition 3.2** (**Communication Overhead Rate**). Let $\Delta W \in \mathbb{R}^{d_1 \times d_2}$ be the update matrix of a model. Suppose the factorization of $\Delta W$ as $\Delta W = AB$, where $A \in \mathbb{R}^{d_1 \times q}$ is a fixed random matrix and $B \in \mathbb{R}^{q \times d_2}$ is a trainable matrix with $q \leq \min(d_1, d_2)$ being the factorization rank. The **communication overhead rate** $\text{CO}_{rate}$ is defined as the ratio of the size of $B$ to the size of $\Delta W$:

$$\text{CO}_{rate} = \frac{\text{size}(B)}{\text{size}(\Delta W)} = \frac{q}{d_1}.$$

**Definition 3.3** (**Reconstruction Error Rate**). Using the same factorization as in Definition 3.2, the **reconstruction error rate** is the expected ratio of the reconstruction error to the original model update. Given full-rank random reconstruction and projection matrices (Assumption 3.1), it is expressed as:

$$\frac{\mathbb{E}_A \left[ \|\Delta W - AB\|_2^2 \right]}{\|\Delta W\|_2^2} = 1 - \frac{q}{d_1}.$$

**Proposition 3.4** (**Single-Vector Factorization**). *Let $\Delta W$, $A$, and $B$ be factorizations of a single layer of the network as in Definition 3.2. By reshaping $\Delta W$ into $\Delta W' \in \mathbb{R}^{d_1 d_2 \times 1}$ the factorization of $\Delta W' = A'B'$ where $A' \in \mathbb{R}^{d_1 d_2 \times p}$ and $B' \in \mathbb{R}^{p \times 1}$ can achieve the same **reconstruction error** and **communication overhead** to the conventional factorization of $\Delta W$ when $p = q d_2$.*

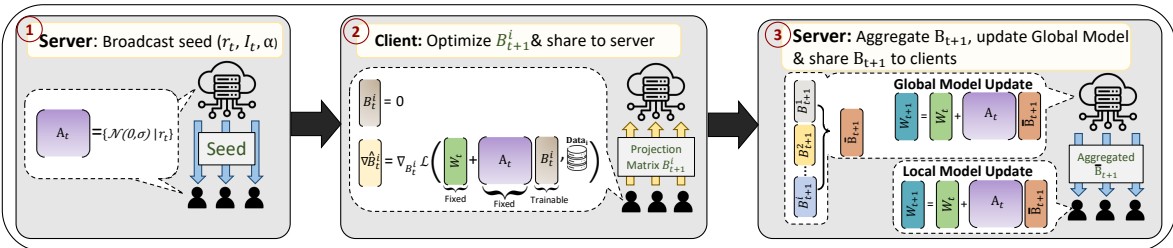

Figure 4: Application of MAPA to communication-efficient FL.

**Proposition 3.5** (**Multi-Layer Factorization**). *Let $\Delta W_i$, $A_i$, and $B_i$ be **single-vector factorization** of $i$-th layer of the $n$-layered network as in Proposition 3.4. By concatenating the reshaped weights $\Delta W_i$ into $\Delta W' \in \mathbb{R}^{d \times 1}$, where $d = \sum_{i=1}^{n} d_1^i d_2^i$. The factorization of $\Delta W' = A'B'$ where $A' \in \mathbb{R}^{d \times p}$ and $B' \in \mathbb{R}^{p \times 1}$ can achieve the same **reconstruction error** and **communication overhead** to the single-vector factorization applied to each $\Delta W_i$ when $p = nq$.*

**Proposition 3.6** (**MAPA Factorization**). *Let $\Delta W$, $A$, $B$, and rank $p$ be multi-layer factorization of a network as defined in Proposition 3.5. By reshaping $\Delta W \in \mathbb{R}^{d \times 1}$ into $\Delta W' \in \mathbb{R}^{\lceil \frac{d}{k} \rceil \times k}$, and the factorization of $\Delta W' = A'B'$ where $A' \in \mathbb{R}^{\lceil \frac{d}{k} \rceil \times 1}$ and $B' \in \mathbb{R}^{1 \times k}$, we can achieve the same **reconstruction error** and **communication overhead** to the multi-layer factorization of $\Delta W$ when $k = p$, while reducing the memory by a factor of $k^2$.*

### 3.2. Application to Communication-Efficient FL

This subsection explains how the factorization outlined in Section 3.1 is utilized in FL, dividing the procedure for clarity. Figure 4 visualizes the outline of this procedure.

**Matrix Construction and Broadcasting.** To ensure consistency across the network, the server and all clients start from an identical condition at each round. We guarantee identical model parameters $W_t$ and reconstruction matrix $A_t$ by broadcasting a random seed $r_t$ and the aggregated projection vector $\bar{B}_t$ at the beginning of round $t$. The initial aggregated projection vector is set to $\bar{B}_0 = \mathbf{0}$.

**In the first round** ($t = 0$), all clients and the server initialize the model $W_0$ using the common seed. The reconstruction matrix $A_0 \in \mathbb{R}^{\frac{d}{k} \times 1}$ is drawn from Gaussian $A \sim \mathcal{N}(0, I)$, and the $i$-th client's local projection matrix $B_0^i \in \mathbb{R}^{1 \times k}$ is set to $0$.

**In subsequent rounds** ($t \geq 1$), clients update their local model $W_t$ using the previous round's matrix $A_{t-1}$, the model parameters $W_{t-1}$, and the broadcasted projection vector $\bar{B}_t$ as follows:

$$W_t = W_{t-1} + A_{t-1}\bar{B}_t. \tag{1}$$

Afterwards, clients generate a new $A_t \sim \mathcal{N}(0, I)$ using the random seed $r_t$ and set $B_t^i \leftarrow \mathbf{0}$. This ensures that $A_t$ and $W_t$ are synchronized and updated.

**Local Optimization of Projection Vector.** This step optimizes the projection vector $\hat{B}_t^i$ that minimizes the local loss function $\mathcal{L}(W_t + A_t B_t^i, \mathcal{D}_i)$, given the random matrix $A_t$. Here, the model weights are derived as $W_t + A_t B_t^i$, and $\mathcal{D}_i$ denotes client $i$'s local dataset.

At each communication round $t \geq 1$, after initializing $A_t$ and $B_t^i$, clients perform local training to optimize $B_t^i$ using their local data $\mathcal{D}_i$. The gradient of the projection vector is computed as:

$$\mathcal{L}_i(W) = \frac{1}{|\mathcal{D}_i|} \sum_{x \in \mathcal{D}_i} \ell(W, x),$$
$$\nabla B_t^i = \nabla_{B_t^i} \mathcal{L}_i(W_t + A_t B_t^i). \tag{2}$$

where $\ell(W, x)$ is the loss function (e.g., cross-entropy loss) computed with model $W$ and data sample $x$.

The optimized projection vector $\hat{B}_t^i$ is then updated using gradient descent:

$$\hat{B}_t^i \leftarrow B_t^i - \eta \nabla B_t^i, \tag{3}$$

where $\eta$ denotes the learning rate. After optimization, clients send their optimized projection vector $\hat{B}_t^i$ to the server. The low dimensionality of $\hat{B}_t^i$ compared to $W_t$ results in communication efficiency.

**Server-Side Aggregation and Global Model Update.** Upon receiving the projection vectors $\hat{B}_t^i$ and their corresponding weights $b_i = |D_i|$ (e.g., batch sizes or number of local samples) from the clients, the server aggregates them to form the global projection vector:

$$\bar{B}_t = \frac{\sum_{i=1}^{N} b_i \hat{B}_t^i}{\sum_{i=1}^{N} b_i}. \tag{4}$$

This weighted averaging captures the collective contribution of all clients, proportional to their data sizes. The server then broadcasts the aggregated projection vector $\bar{B}_t$ to all clients. After receiving $\bar{B}_t$, the server and all clients update their local models using the reconstruction matrix $A_t$ and the aggregated projection vector $\bar{B}_t$ as:

$$W_{t+1} = W_t + A_t \bar{B}_t. \tag{5}$$

This update integrates the clients' optimized directions into their local models and ensures synchronization across the network. This process is repeated until the global model converges. Abbreviated pseudo-code is provided in Algorithm 1, while Appendix A offers a more detailed version.

Table 1: Summary of datasets and models used in our experiments.

| Dataset | Client Distribution | Train/Test | # Classes | Model | # Parameters |
|---------|--------------------|-----------|-----------|-------|--------------|
| MNIST | Non-IID (2 classes) | 60K / 10K | 10 | CNN - 2 Layers | 11,274 |
| FMNIST | Non-IID (2 classes) | 60K / 10K | 10 | CNN - 2 Layers | 11,274 |
| CIFAR-10 | Non-IID (2 classes) | 50K / 10K | 10 | CNN - 4 Layers | 1,146,634 |
| CIFAR-100 | Non-IID (10 classes) | 50K / 10K | 100 | WideResNet 16d4w | 2,854,420 |
| TinyImageNet | Non-IID (10 classes) | 100K / 10K | 200 | WideResNet 16d4w | 2,880,120 |
| Shakespeare | Distributed by Roles | 14K / 2K | 65 | LSTM | 814,957 |
| Sentiment140 | Distributed by Users | 1.4M / 200K | 2 | Transformer | 2,221,570 |

---

**Algorithm 1** FL with MAPA

**Initialize:** Global model $W_0 \in \mathbb{R}^{\lceil \frac{d}{k} \rceil \times 1}$, reconstruction matrix $A_0 \in \mathbb{R}^{\lceil \frac{d}{k} \rceil \times 1}$, projection matrix $\bar{B}_0 \leftarrow \mathbf{0} \in \mathbb{R}^{1 \times k}$, seed $r_0$

1: **for** each communication round $t = 1, \ldots, T - 1$ **do**
2:    **Server:** Broadcast global $\bar{B}_{t-1}$ and $r_{t-1}$
3:    **for** each client $i = 1, \ldots, N$ **in parallel do**
4:       **Client:** Receive $\bar{B}_{t-1}$ and $r_{t-1}$
5:       Update $W_t = W_{t-1} + A_{t-1}\bar{B}_{t-1}$
6:       Update $A_t = \mathcal{N}(0, \sigma)|r_{t-1}$
7:       Initialize $B_t^i \leftarrow \mathbf{0} \in \mathbb{R}^{1 \times k}$
8:       **for** each local epoch $e = 1, \ldots, E$ **do**
9:          $\nabla B_t^i = \nabla_{B_t^i} \mathcal{L}_i(W_t + A_t B_t^i, \mathcal{D}_i)$
10:          Update $\hat{B}_t^i \leftarrow B_t^i - \eta \nabla B_t^i$
11:       **end for**
12:       Send updated $\hat{B}_t^i$ to server
13:    **end for**
14:    **Server:** Aggregate $\bar{B}_t \leftarrow \frac{1}{S} \sum_{i=1}^{N} b_i \hat{B}_t^i$
15:    Update global model $W_{t+1} \leftarrow W_t + A_t \bar{B}_t$
16:    Update random seed $r_t$
17: **end for**
18: **Return:** Final global model $W_T$

---

## 4. Convergence Analysis

We examine the convergence dynamics of FL with MAPA.

**Assumption 4.1.** For each $i, \mathcal{L}_i(v)$ is $\beta$-smooth, i.e., $\|\nabla \mathcal{L}_i(u) - \nabla \mathcal{L}_i(v)\| \leq \beta \|u - v\|$ for any $u, v$.

**Assumption 4.2.** Variance of the stochastic gradient of $D_i$ is bounded for each client $i$, i.e.,

$$\mathbb{E}\left[\left\|\nabla \mathcal{L}_i(W) - \widetilde{\nabla} \mathcal{L}_i(W)\right\|^2\right] \leq \sigma_l^2$$

.

**Theorem 4.3.** *Let the learning rate satisfy* $\eta_t \leq \frac{1-4\epsilon}{4\beta(1+\epsilon)}$. *Then, the algorithm achieves the following bound:*

$$\frac{1}{4H_T} \sum_{t=0}^{T-1} \eta_t \mathbb{E}\left[\|\nabla \mathcal{L}(W_t)\|^2\right] \leq$$

$$\frac{\mathbb{E}\left[\mathcal{L}(W_0)\right] - \mathcal{L}^*}{H_T} + 2(\epsilon + \beta + \beta\epsilon)\sigma_l^2 \frac{1}{H_T} \sum_{t=0}^{T-1} \eta_t^2,$$

*where* $H_T = \sum_{t=0}^{T-1} \eta_t$, $\epsilon$ *is the distortion parameter from the JL Lemma, and* $\mathcal{L}^*$ *is the minimum value of* $\mathcal{L}(W)$.

With a decreasing learning rate satisfying $\sum_{t=0}^{\infty} \eta_t \to \infty$, $\sum_{t=0}^{\infty} \eta_t^2 < \infty$ ($\eta_t = \frac{\eta_0}{t+c}$ for some constants $\eta_0 > 0$, $c > 0$), the term $H_T = \sum_{t=0}^{T-1} \eta_t$ grows unbounded, while the weighted sum $\sum_{t=0}^{T-1} \eta_t^2$ remains finite. Therefore, the right-hand side of Theorem 4.3's bound satisfies:

$$\frac{\mathbb{E}[\mathcal{L}(W_0)] - \mathcal{L}^*}{H_T} \to 0, \quad \frac{1}{H_T} \sum_{t=0}^{T-1} \eta_t^2 \to 0 \quad \text{as } T \to \infty.$$

Thus, the gradient norm average satisfies:

$$\frac{1}{H_T} \sum_{t=0}^{T-1} \eta_t \mathbb{E}\left[\|\nabla \mathcal{L}(W_t)\|^2\right] \to 0,$$

confirming convergence to a stationary point.

As shown above, the convergence bound is influenced by the factor $\epsilon + \beta + \beta\epsilon$. In particular, the bound becomes tightest and achieves the highest communication efficiency when there is no reconstruction error, i.e., when $\epsilon = 0$. The complete proof of Theorem 4.3 is located in Appendix D, and Appendix F contains notation table used in this work.

## 5. Experimental Setup

MAPA is evaluated across multiple model architectures, tasks, and baselines. The benchmarks include five image classification tasks, next-character prediction, and sentiment analysis. For image classification, we use MNIST (Le-Cun et al., 1998), FMNIST (Xiao et al., 2017), CIFAR-10, CIFAR-100 (Krizhevsky et al., 2009), and TinyImageNet (University, 2015). For sequential tasks, we use Shakespeare and Sentiment140 from the LEAF dataset (Caldas et al., 2018), which is specifically designed for FL scenarios. The details of each dataset and the model architectures used for training are summarized in Table 1, demonstrating MAPA's adaptability across varying scales and models.

**Non-IID Distribution.** To simulate realistic FL conditions, we distribute the training datasets non-IID across 100 clients. For image classification, we assign each client a unique subset of classes. For NLP tasks, we follow the natural partitioning of the LEAF dataset, where different Shakespearean roles and Twitter users represent individual clients. Model performance is evaluated using the original test sets.

**Model Architectures.** To evaluate MAPA's scalability, we experiment with models of varying sizes. A 2-layer CNN

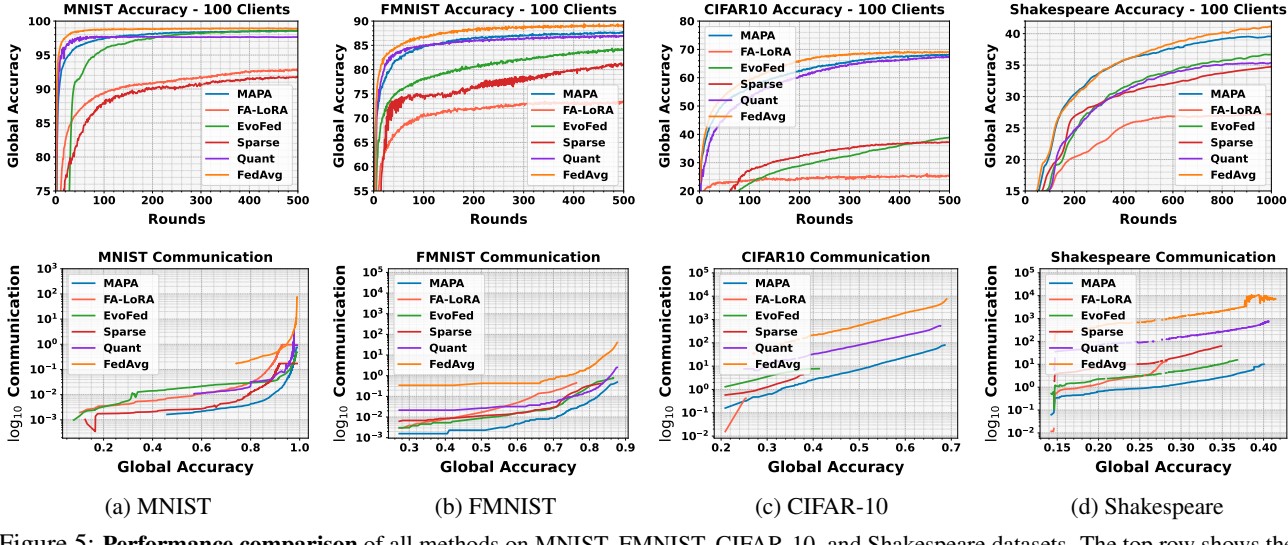

| (a) MNIST | (b) FMNIST | (c) CIFAR-10 | (d) Shakespeare |

Figure 5: **Performance comparison** of all methods on MNIST, FMNIST, CIFAR-10, and Shakespeare datasets. The top row shows the accuracy for the respective datasets, while the bottom row illustrates the communication cost associated with each level of accuracy.

Table 2: Summary of performance and communication cost in non-IID setting.

| | MNIST | | FMNIST | | CIFAR-10 | | CIFAR-100 | | Shakespeare | | Sent140 | | TinyImageNet | |
| --- | --- | --- | --- | --- | --- | --- | --- | --- | --- | --- | --- | --- | --- | --- |
| | Com. | Acc. | Com. | Acc. | Com. | Acc. | Com. | Acc. | Com. | Acc. | Com. | Acc. | Com. | Acc. |
| FedAvg | 100% | 98.9% | 100% | 89.2% | 100% | 69.0% | 100% | 43.47% | 100% | 41.86% | 100% | 74.90% | 100% | 36.48% |
| Sparse | 15.3% | 92.1% | 24.1% | 81.1% | 2.7% | 37.15% | 1.20% | 33.72% | 1.73% | 34.86% | 1.93% | 74.21% | 1.32% | 25.34% |
| Quantize | 31.3% | 97.6% | 24.1% | 87.1% | 15.2% | 67.40% | 6.10% | 40.05% | 10.11% | 35.45% | 13.85% | 73.70% | 8.75% | 34.47% |
| EvoFed | 9.40% | **98.5%** | 7.60% | 84.7% | 3.4% | 39.50% | 20.4% | 37.62% | 0.23% | 36.76% | 0.40% | 70.50% | 1.85% | 15.4% |
| FA-LoRA | 30.2% | 93.8% | 17.9% | 74.1% | 1.7% | 23.52% | 1.20% | 19.10% | 1.67% | 28.07% | 1.30% | 66.61% | 1.27% | 7.31% |
| **MAPA** | **2.90%** | **98.5%** | **3.10%** | **88.0%** | **1.20%** | **68.3%** | **0.91%** | **40.16%** | **0.13%** | **39.96%** | **0.19%** | **74.50%** | **0.97%** | **35.22%** |

is used for MNIST and FMNIST, while a 4-layer CNN is employed for CIFAR-10. For larger datasets, including CIFAR-100 and TinyImageNet, we use WideResNet variants with 4 widths and 16 depths. Beyond CNNs, we extend our evaluation to LSTM for next-character prediction and Transformers for sentiment analysis, ensuring MAPA's effectiveness among various networks. Appendix E includes the details of the model architectures and hyperparameters.

**Baselines.** We compare the proposed MAPA with several baselines, including FedAvg, FedAvg with Sparsification (Sparse), and FedAvg with Quantization (Quant), as common compression techniques. Additionally, we evaluate MAPA against EvoFed (Rahimi et al., 2024), a state-of-the-art compression-based method, and FA-LoRA, inspired by (Sun et al., 2024; Zhang et al., 2023; Zhu et al., 2024; Hao et al., 2024), as a representative factorization-based approaches. Comparisons with Sparsification and Quantization establish MAPA's effectiveness relative to standard compression methods. EvoFed serves as a strong baseline designed for FL and gradient compression, demonstrating how MAPA's subspace optimization can surpass existing SOTA of communication-efficient FL techniques that apply compression post-optimization. Finally, the evaluation of MAPA alongside FA-LoRA highlights the impact of MAPA's dynamic subspace exploration in improving the convergence and performance of LoRA-based techniques.

**Federated Learning Setting.** In each training round, 10% of the clients are randomly sampled to participate. These selected clients train in parallel and transmit only updates to the server. The server then aggregates updates and returns it to the clients. Model performance on the test dataset and communication load per client is evaluated at the server.

## 6. Results and Discussions

We discuss the experimental results in detail and provide further insights into the MAPA's performance. The accuracy of MAPA, compared with multiple baseline methods and different datasets, is shown in Figure 5 (top row). MAPA outperforms all other methods in all tasks and delivers results comparable to FedAvg while utilizing a much smaller number of trainable parameters due to the promotion of exploration and effective utilization of the communication budget for reducing the loss function directly. Figure 5 (bottom-row) shows each method's minimum amount of communication to reach any accuracy. It can be seen that MAPA tends to utilize significantly less communication than other techniques, as the communication cost (y-axis) is in the $\log_{10}$ scale. The additional results for CIFAR100, TinyImagenet, and Sentiment140 are placed in Appendix B.

Table 2 summarizes experimental results by showing the maximum accuracy of each baseline and the communication cost percentage compared to FedAvg for reaching a certain

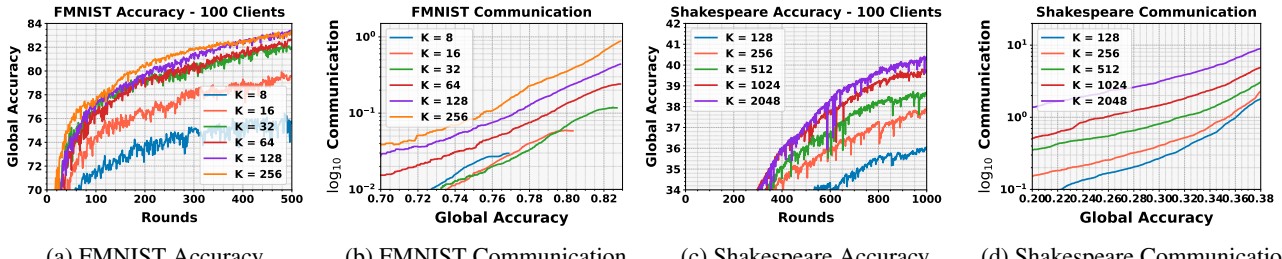

(a) FMNIST Accuracy     (b) FMNIST Communication     (c) Shakespeare Accuracy     (d) Shakespeare Communication

Figure 6: Accuracy and communication cost per accuracy level for FMNIST and Shakespeare dataset. Demonstrating the effect of a number of trainable parameters ($k$) on the communication efficiency of MAPA.

performance. This certain performance is selected as the maximum accuracy of the worst baseline, so all models are fairly compared on how much communication they need to reach the same accuracy. MAPA achieves significantly lower communication costs than FedAvg while maintaining competitive accuracy levels. In MNIST and FMNIST datasets, MAPA achieves 99.6% and 98.6% of FedAvg accuracy while having only 3% of communication. Similarly, in CIFAR-10, CIFAR-100, and TinyImagenet datasets, it reaches 98.9%, 92.4%, and 96.5% of FedAvg accuracy with around 1.0% of communication. Finally, in Shakespeare and Sentiment140, We see that it preserves up to 95.5% and 99.5% of FedAvg's accuracy while significantly cutting communication to lower than 0.2% of FedAvg.

**MAPA Hyperparameter.** MAPA streamlines the LoRA approach by applying a single factorization to the entire model's parameters, eliminating the need to fine-tune and optimize factorization configurations for each layer. Instead, MAPA enables performance control through a single parameter, $k$, which directly influences both communication cost and model accuracy. This section examines the effect of varying $k$ on model performance and communication efficiency in FL. Figure 6 presents results for the FMNIST and Shakespeare datasets. As expected, smaller $k$ values reduce communication costs but slow convergence, often requiring significantly more training rounds. Conversely, larger $k$ values exponentially increase communication overhead while yielding diminishing performance gains. To strike an optimal balance, we aim for a $k$ that meets a desired performance threshold with minimal total communication. In some cases, increasing $k$ accelerates convergence to higher accuracy, ultimately improving communication efficiency. Subfigures (b) and (c) of Figure 6 illustrate the communication overhead required to reach specific accuracy, providing a basis for selecting the most communication-efficient $k$. A similar approach is used to determine the optimal parameter settings for other baselines, ensuring a fair comparison.

**Fresh Reconstruction Matrix.** A key factor in MAPA's superiority over FA-LoRA is its use of a dynamically generated reconstruction matrix $A$ rather than a fixed one. This approach promotes the exploration of new subspaces throughout training. Figure 7 illustrates the benefits of using a fresh

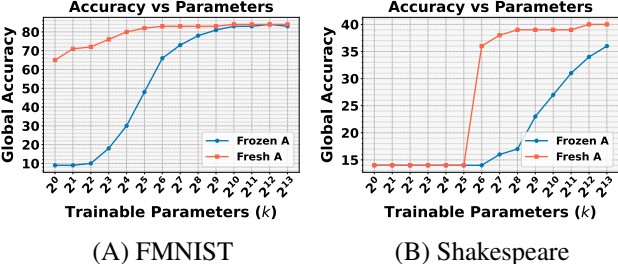

(A) FMNIST      (B) Shakespeare

Figure 7: Comparison of having a fresh $A$ vs. frozen $A$.

$A$ on the FMNIST and Shakespeare datasets. We evaluate MAPA across varying numbers of trainable parameters, ranging from $2^0$ to $2^{13}$. For FMNIST, this corresponds to 0.009% to 72.27% of the total model parameters, while for Shakespeare, it spans from 0.0001% to nearly 1%. In both cases, MAPA with a fresh $A$ achieves superior convergence with fewer parameters, effectively leveraging the search space. In contrast, when $A$ is frozen, performance follows a logarithmic correlation with the number of trainable parameters, requiring an exponentially larger parameter count to match the results obtained with a fresh $A$.

**Additional Results.** Appendix G supplements our experiments with additional evaluations on IID distribution and the absence of client sampling. Furthermore, Appendix H presents a memory complexity analysis, emphasizing the computational efficiency and flexibility of MAPA compared to layer-wise low-rank factorization.

## 7. Conclusion

We introduced *Model-Agnostic Projection Adaptation*, a novel approach to communication-efficient FL. Unlike layer-wise LoRA, MAPA factorizes the entire model's parameters into a compact projection vector and a randomly regenerated reconstruction matrix, enabling efficient updates without architecture-specific constraints and mitigating FA-LoRA suboptimality while flexibly balancing communication and accuracy. Our theoretical analysis establishes MAPA's convergence, and extensive experiments demonstrate its superiority over existing compression and LoRA-based methods across diverse datasets. MAPA significantly reduces communication while maintaining strong performance, making it a practical and scalable solution for FL.

## Impact Statement

This paper presents work aimed at advancing the field of Federated Learning by improving communication efficiency in distributed training. The proposed Model-Agnostic Projection Adaptation (MAPA) method reduces the communication overhead while maintaining model performance, making federated learning more practical for large-scale and resource-constrained environments. By enabling efficient model training without centralizing user data, MAPA supports privacy-preserving AI applications in areas such as healthcare, finance, and edge computing. However, as with all federated learning methods, potential societal impacts include challenges related to fairness, client participation incentives, and susceptibility to adversarial attacks. Future work should consider these aspects to ensure equitable and secure deployment. Overall, MAPA contributes positively to the scalability and accessibility of federated learning, with no immediate ethical concerns requiring specific attention.

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

# A. Full Pseudocode for Federated Learning with MAPA

---

**Algorithm 2** Federated Learning with MAPA (Detailed Version)

---

1: **Initialization:**
2:  - Seed all clients and the server with the same initial random seed $r_0$.
3:  - Initialize the global model, reconstruction matrix, and projection vector:

$$W_0 \in \mathbb{R}^{\lceil \frac{d}{k} \rceil \times k}, \quad \text{e.g. drawn from } \mathcal{N}(0, \sigma^2 I_d) \text{ or any other scheme.}$$

$$A_0 \in \mathbb{R}^{\lceil \frac{d}{k} \rceil \times 1}, \quad \text{with each column drawn i.i.d. from } \mathcal{N}(0, \sigma^2 I_d).$$

$$\bar{B}_0 \leftarrow \mathbf{0} \in \mathbb{R}^{1 \times k}.$$

4: **for** each communication round $t = 1, \ldots, T - 1$ **do**
5:  **On the Server:**
6:    1. Broadcast the current global projection vector $\bar{B}_{t-1}$ and the current PRNG seed $r_{t-1}$ to all clients.
7:  **On Each Client** $i = 1, \ldots, N$ **(in parallel):**
8:    2. Receive $\bar{B}_{t-1}$ and $r_{t-1}$.
9:    3. Update the local model: $$W_t \leftarrow W_{t-1} + A_{t-1} \bar{B}_{t-1}.$$

10:    4. Re-generate the reconstruction matrix using the seed $r_{t-1}$:

$$A_t = \mathcal{N}(0, \sigma^2 I_d)\big|_{r_{t-1}}.$$

11:       (This means each client and the server can reproduce $A_t$ identically using the same seed.)
12:    5. Initialize the local projection vector: $$B_t^i \leftarrow \mathbf{0} \in \mathbb{R}^{1 \times k}.$$

13:    6. Perform local training for $E$ epochs (or mini-batch steps). For each local epoch $e = 1, \ldots, E$:
14:       (a) Compute the gradient of the local loss $\mathcal{L}_\rangle$ w.r.t. the projection vector $B_t^i$:

$$\nabla B_t^i \ = \ \nabla_{B_t^i} \mathcal{L}_\rangle \big( W_t + A_t B_t^i, \ \mathcal{D}_i \big).$$

15:       (b) Update the local projection vector with your choice of optimizer (e.g., SGD):

$$\hat{B}_t^i \ \leftarrow \ B_t^i - \eta \, \nabla B_t^i.$$

16:       (c) Optionally set $B_t^i \leftarrow \hat{B}_t^i$ if doing iterative local steps.
17:    7. Send the locally updated projection vector $\hat{B}_t^i$ back to the server.
18:  **On the Server (after all clients respond):**
19:    8. (Re)generate $A_t$ with the seed $r_{t-1}$ so the server is consistent with clients:

$$A_t = \mathcal{N}(0, \sigma^2 I_d)\big|_{r_{t-1}}.$$

20:    9. Aggregate the projection vectors. Let $b_i$ be any weighting factor for client $i$, or just set $b_i = 1$ if unweighted:

$$\bar{B}_t \ \leftarrow \ \frac{1}{S} \sum_{i=1}^{N} b_i \, \hat{B}_t^i,$$

where $S = \sum_{i=1}^{N} b_i$.
21:    10. Update the global model: $$W_{t+1} \ \leftarrow \ W_t + A_t \bar{B}_t.$$

22:    11. Generate a new random seed $r_t$ (e.g., $r_t = \text{hash}(r_{t-1})$).
23: **end for**
24: **Return:** The final global model $W_T$.

---

# B. Accuracy and Communication Learning curves

The common practice of implementing matrix factorization in communication-efficient FL involves using a fixed and frozen reconstruction matrix throughout the whole training. In contrast, we found that having a reconstructed matrix generated fresh and independently in each round outperforms this traditional choice without any additional communication overhead. Figure 8 shows the evidence of this improvement in the case of FMNIST training with 100 clients.

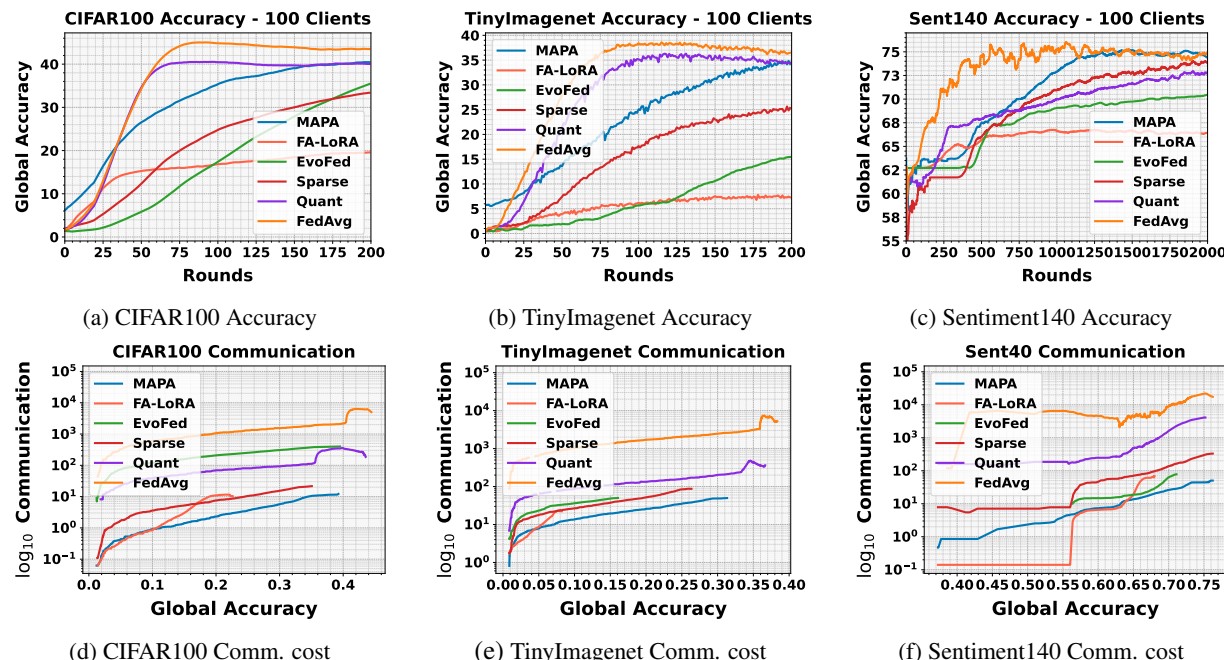

(a) CIFAR100 Accuracy        (b) TinyImagenet Accuracy        (c) Sentiment140 Accuracy

(d) CIFAR100 Comm. cost        (e) TinyImagenet Comm. cost        (f) Sentiment140 Comm. cost

Figure 8: **Performance comparison** of MAPA and baseline methods on CIFAR100, TinyImagenet, and Sentiment140 datasets. The top row shows the accuracy achieved by each method on the respective datasets, while the bottom row illustrates the communication cost associated with each method.

# C. Proof of Definitions and Propositions

**Definition C.1** (**Communication Overhead Rate**). Let $\Delta W \in \mathbb{R}^{d_1 \times d_2}$ be the update matrix of a model. Suppose the factorization of $\Delta W$ as $\Delta W = AB$, where $A \in \mathbb{R}^{d_1 \times q}$ is a fixed random matrix and $B \in \mathbb{R}^{q \times d_2}$ is a trainable matrix with $q \leq \min(d_1, d_2)$ being the factorization rank. The **communication overhead rate** $\text{CO}_{rate}$ is defined as the ratio of the size of $B$ to the size of $\Delta W$:

$$\text{CO}_{rate} = \frac{\text{size}(B)}{\text{size}(\Delta W)} = \frac{q}{d_1}.$$

**Definition C.2** (**Reconstruction Error Rate**). Using the same factorization as in Definition 3.2, the **reconstruction error rate** is the expected ratio of the reconstruction error to the original model update. Given full-rank random reconstruction and projection matrices (Assumption 3.1), it is expressed as:

$$\frac{\mathbb{E}_A \left[ \|\Delta W - AB\|_2^2 \right]}{\|\Delta W\|_2^2} = 1 - \frac{q}{d_1}.$$

*Proof.* Let $\Delta W = [\Delta w_1 \ \Delta w_2 \ \cdots \ \Delta w_{d_2}]$, where each column $\Delta w_i \in \mathbb{R}^{d_1}$. Similarly, the reconstruction $AB$ can be written as $[Ab_1 \ Ab_2 \ \cdots \ Ab_{d_2}]$, where each $b_i \in \mathbb{R}^q$ is a trainable matrix.

The reconstruction error is given by:

$$\|\Delta W - AB\|_2^2 = \sum_{i=1}^{d_2} \|\Delta w_i - Ab_i\|_2^2.$$

The projection of $\Delta w_i$ onto the subspace spanned by $A$ is $P_A \Delta w_i$. The error rate $E$ is defined as:

$$E = \frac{\|\Delta w_i - P_A \Delta w_i\|_2^2}{\|\Delta w_i\|_2^2}.$$

Using the Pythagorean theorem:

$$\|\Delta w_i\|_2^2 = \|P_A \Delta w_i\|_2^2 + \|w_i - P_A \Delta w_i\|_2^2,$$

we rewrite $E$ as:

$$E = \frac{\|\Delta w_i\|_2^2 - \|P_A \Delta w_i\|_2^2}{\|\Delta w_i\|_2^2} = 1 - \frac{\|P_A \Delta w_i\|_2^2}{\|\Delta w_i\|_2^2}.$$

The expected value of $\|P_A \Delta w_i\|_2^2$ for a full-rank random Gaussian projection is:

$$\mathbb{E}[\|P_A \Delta w_i\|_2^2] = \frac{q}{d_1} \|\Delta w_i\|_2^2.$$

Substituting this into $E$:

$$\mathbb{E}[\|\Delta w_i - Ab_i\|_2^2] = 1 - \frac{\mathbb{E}[\|P_A \Delta w_i\|_2^2]}{\|\Delta w_i\|_2^2} = 1 - \frac{\frac{p}{d} \|\Delta w_i\|_2^2}{\|w_i\|_2^2} = 1 - \frac{q}{d_1}.$$

Applying this to each column $\Delta \Delta w_i$ of $\Delta W$, we obtain:

$$\mathbb{E}_A \left[ \sum_{i=1}^{d_2} \|\Delta w_i - Ab_i\|_2^2 \right] = \sum_{i=1}^{d_2} \mathbb{E}_A \left[ \|\Delta w_i - P_A(\Delta w_i)\|_2^2 \right].$$

Using the expected error formula:

$$= \sum_{i=1}^{d_2} \left( 1 - \frac{q}{d_1} \right) \|\Delta w_i\|_2^2 = \left( 1 - \frac{q}{d_1} \right) \sum_{i=1}^{d_2} \|\Delta w_i\|_2^2.$$

Since $\|\Delta W\|_2^2 = \sum_{i=1}^{d_2} \|\Delta w_i\|_2^2$, we get:

$$\mathbb{E}_A \left[ \|\Delta W - AB\|_2^2 \right] = \left( 1 - \frac{q}{d_1} \right) \|\Delta W\|_2^2.$$

$\square$

**Proposition C.3** (**Single-Vector Factorization**). *Let $\Delta W$, $A$, and $B$ be factorizations of a single layer of the network as in Definition 3.2. By reshaping $\Delta W$ into $\Delta W' \in \mathbb{R}^{d_1 d_2 \times 1}$ the factorization of $\Delta W' = A'B'$ where $A' \in \mathbb{R}^{d_1 d_2 \times p}$ and $B' \in \mathbb{R}^{p \times 1}$ can achieve the same **reconstruction error** and **communication overhead** to the conventional factorization of $\Delta W$ when $p = qd_2$.*

*Proof.* **Error Preservation.** In the single-vector setup, $\Delta W' \in \mathbb{R}^{d_1 d_2}$ is projected onto a subspace of dimension $p$. From random projection theory (as used in Definition 3.3), if $A'$ is sampled such that $\mathrm{rank}(A') = p$, then:

$$\mathbb{E}\left[ \frac{\|\Delta W' - A'B'\|_2^2}{\|\Delta W'\|_2^2} \right] = 1 - \frac{p}{d_1 d_2}.$$

Substituting $p = qd_2$ gives:

$$1 - \frac{qd_2}{d_1 d_2} = 1 - \frac{q}{d_1}.$$

Hence, the expected reconstruction error satisfies:

$$\mathbb{E}\left[ \|\Delta W' - A'B'\|_2^2 \right] = \left( 1 - \frac{q}{d_1} \right) \|\Delta W'\|_2^2,$$

which matches the original factorization.

**Communication Overhead Preservation.** Since $\Delta W' \in \mathbb{R}^{d_1 d_2}$, its total size is $\mathrm{size}(\Delta W') = d_1 d_2$. For the new factorization, we have:

$$\mathrm{size}(B') = p = qd_2.$$

Thus, the communication overhead is:

$$\mathrm{CO}'_{rate} = \frac{\mathrm{size}(B')}{\mathrm{size}(\Delta W')} = \frac{qd_2}{d_1 d_2} = \frac{q}{d_1}.$$

which matches the original overhead.

Since both the expected reconstruction error and the communication overhead remain unchanged, the single-vector factorization with $p = qd_2$ is equivalent in terms of efficiency. $\square$

**Proposition C.4** (**Multi-Layer Factorization**). *Let $\Delta W_i$, $A_i$, and $B_i$ be **single-vector factorization** of $i$-th layer of the $n$-layered network as in Proposition 3.4. By concatenating the reshaped weights $\Delta W_i$ into $\Delta W' \in \mathbb{R}^{d \times 1}$, where $d = \sum_{i=1}^{n} d_1^i d_2^i$. The factorization of $\Delta W' = A'B'$ where $A' \in \mathbb{R}^{d \times p}$ and $B' \in \mathbb{R}^{p \times 1}$ can achieve the same **reconstruction error** and **communication overhead** to the single-vector factorization applied to each $\Delta W_i$ when $p = nq$.*

*Proof.* **Error Preservation.** For each layer $i$, a random full-rank matrix $A_i \in \mathbb{R}^{d_1^i \times q}$ yields an expected squared reconstruction error

$$\mathbb{E}\left[ \|\Delta W_i - A_i B_i\|_F^2 \right] = \left( 1 - \frac{q}{d_1^i} \right) \|\Delta W_i\|_F^2.$$

Flattening $\Delta W_i$ into $\Delta W'_i \in \mathbb{R}^{(d_1^i d_2^i) \times 1}$, a single-vector projection of dimension $q\, d_2^i$ preserves this same error ratio (cf. Proposition 3.4).

When we concatenate all $\Delta W'_i$ into $\Delta W' \in \mathbb{R}^{d \times 1}$, we form a block-structured vector. Let $p := n q$ and let $A' \in \mathbb{R}^{d \times p}$ be constructed from Gaussian distribution. By the standard random-projection argument in dimension $d$ with subspace size $p$,

$$\mathbb{E}\left[ \|\Delta W' - A'B'\|_2^2 \right] = \left( 1 - \frac{p}{d} \right) \|\Delta W'\|_2^2.$$

Since $p = n q$, the overall ratio matches applying single-vector factorizations of rank $q$ to each $\Delta W'_i$ individually.

**Communication Overhead Preservation.** For each layer $i$, the single-vector factorization of $\Delta W_i$ introduces

$$\mathrm{size}(B_i) = q\, d_2^i, \quad \mathrm{size}(\Delta W_i) = d_1^i\, d_2^i, \quad \text{hence} \quad \frac{\mathrm{size}(B_i)}{\mathrm{size}(\Delta W_i)} = \frac{q}{d_1^i}.$$

Concatenating all $\Delta W_i'$ into $\Delta W' \in \mathbb{R}^{d \times 1}$ gives $\text{size}(\Delta W') = d$, with

$$d \;=\; \sum_{i=1}^{n} d_1^i \, d_2^i.$$

Meanwhile, in the multi-layer factorization, the new trainable vector $B' \in \mathbb{R}^{p \times 1}$ has

$$\text{size}(B') \;=\; p \;=\; n \, q.$$

Thus

$$\frac{\text{size}(B')}{\text{size}(\Delta W')} \;=\; \frac{n \, q}{\sum_{i=1}^{n} \left( d_1^i \, d_2^i \right)},$$

which matches the *total* overhead of $n$ individual rank-$q$ factorizations (one per layer) in aggregate. Consequently, the communication overhead rate is also preserved.

Since both the expected reconstruction error (per layer or in total) and the communication overhead remain the same, choosing $p = n \, q$ for $\Delta W'$ is equivalent to applying single-vector factorization of rank $q$ separately to each layer. □

**Proposition C.5** (**MAPA Factorization**). *Let $\Delta W$, $A$, $B$, and rank $p$ be multi-layer factorization of a network as defined in Proposition 3.5. By reshaping $\Delta W \in \mathbb{R}^{d \times 1}$ into $\Delta W' \in \mathbb{R}^{\lceil \frac{d}{k} \rceil \times k}$, and the factorization of $\Delta W' = A'B'$ where $A' \in \mathbb{R}^{\lceil \frac{d}{k} \rceil \times 1}$ and $B' \in \mathbb{R}^{1 \times k}$, we can achieve the same **reconstruction error** and **communication overhead** to the multi-layer factorization of $\Delta W$ when $k = p$, while reducing the memory by a factor of $k^2$.*

*Proof.* **Error Preservation.** Since $\Delta W \in \mathbb{R}^{d \times 1}$ is reshaped into $\Delta W' \in \mathbb{R}^{\lceil d/k \rceil \times k}$, we still have $\|\Delta W'\|_F^2 = \|\Delta W\|_2^2$. When $A' \in \mathbb{R}^{\lceil d/k \rceil \times 1}$ is a suitable random projection (and $B' \in \mathbb{R}^{1 \times k}$ is fit accordingly), the rank-1 subspace of dimension 1 within $\lceil d/k \rceil$ induces the known expected error ratio

$$\mathbb{E}\Big[\|\Delta W' - A'B'\|_F^2\Big] \;=\; \Big(1 - \tfrac{1}{\lceil d/k \rceil}\Big) \|\Delta W'\|_F^2,$$

since the ambient dimension is $\lceil d/k \rceil \times k \approx d$. By taking $k = p$, we obtain (via standard random-projection arguments) the matching error ratio $1 - p/d$, up to negligible rounding. Therefore:

$$\mathbb{E}\Big[\|\Delta W' - A'B'\|_F^2\Big] \;=\; \Big(1 - \frac{p}{d}\Big) \|\Delta W'\|_F^2,$$

**Communication Overhead Preservation.** The matrix $B' \in \mathbb{R}^{1 \times k}$ has size $k$ in total. Meanwhile, $\Delta W' \in \mathbb{R}^{\lceil d/k \rceil \times k}$ has size $\lceil d/k \rceil \, k \approx d$. Thus

$$\frac{\text{size}(B')}{\text{size}(\Delta W')} \;=\; \frac{k}{\lceil d/k \rceil \, k} \;\approx\; \frac{k}{d} = \frac{p}{d}.$$

Setting $k = p$ matches the original ratio $\frac{p}{d}$ from $B \in \mathbb{R}^{p \times 1}$ in the multi-layer factorization.

**Memory Reduction by Factor $k^2$.** In standard rank-$p$ factorizations for $\Delta W \in \mathbb{R}^{d \times 1}$, one typically stores a $d \times p$ projection plus a $p \times 1$ vector, whose total size scales as $dp + p$. By contrast, $A' \in \mathbb{R}^{\lceil d/k \rceil \times 1}$ plus $B' \in \mathbb{R}^{1 \times k}$ has combined size $\lceil d/k \rceil + k$. When $k = p$, the ratio of these sizes can be shown to drop by a factor of approximately $k^2$. Hence the approach allocates $k^2$ times less memory than a naive $d \times p$ plus $p \times 1$ arrangement. As $p = k$

$$\frac{dp + p}{\lceil d/k \rceil + k} \;=\; \frac{dk + k}{\lceil d/k \rceil + k} \;\approx\; \frac{d + 1}{d/k^2 + 1} \;\approx\; k^2$$

Thus, the factorization $\Delta W' = A'B'$ with $k = p$ exactly preserves the original rank-$p$ error and overhead while using $k^2$-fold less memory. □

## D. Proof of Theorem

### D.1. Assumptions and Preliminaries

We restate the key assumptions required for the convergence analysis.

**Assumption D.1** (Smoothness). For each $i$, $\mathcal{L}_i(W)$ is $\beta$-smooth, i.e.,

$$\|\nabla \mathcal{L}_i(u) - \nabla \mathcal{L}_i(v)\| \leq \beta \|u - v\|, \quad \text{for all } u, v.$$

**Assumption D.2** (Bounded Variance of Stochastic Gradients). The variance of the stochastic gradient estimator $\widetilde{\nabla} \mathcal{L}_i(W_t)$ is bounded, i.e., $\mathbb{E}\left[\left\|\widetilde{\nabla} \mathcal{L}_i(W_t) - \nabla \mathcal{L}_i(W_t)\right\|^2\right] \leq \sigma_l^2$, for all clients $i$ and iterations $t$.

**Lemma D.3** (Johnson-Lindenstrauss Lemma). *Given $0 < \epsilon < 1$, a set of points $\{x_1, x_2, \ldots, x_N\} \subset \mathbb{R}^d$, and a target dimension $k = O\left(\frac{\log N}{\epsilon^2}\right)$, there exists a random linear mapping $P \in \mathbb{R}^{k \times d}$ such that for all $i, j$:*

$$(1 - \epsilon)\|x_i - x_j\|^2 \leq \|P x_i - P x_j\|^2 \leq (1 + \epsilon)\|x_i - x_j\|^2.$$

In our context, the random projection matrices $B_t^i$ and reconstruction matrices $A_t$ satisfy the JL property with high probability.

### D.2. Proof of Theorem 1

**Theorem D.1.** *Given a decreasing learning rate $\eta_t \leq \frac{1 - 4\epsilon}{4\beta(1 + \epsilon)}$, the algorithm has the following convergence bound:*

$$\frac{1}{4 H_T} \sum_{t=0}^{T-1} \eta_t \mathbb{E}\left[\|\nabla \mathcal{L}(W_t)\|^2\right] \leq \frac{\mathbb{E}\left[\mathcal{L}(W_0)\right] - \mathcal{L}^*}{H_T} + 2(\epsilon + \beta + \beta\epsilon)\sigma_l^2 \left(\frac{1}{H_T} \sum_{t=0}^{T-1} \eta_t^2\right)$$

*where $H_T = \sum_{t=0}^{T-1} \eta_t$, $\epsilon$ is the distortion parameter from the JL Lemma, and $\mathcal{L}^*$ represents the minimum value of $\mathcal{L}(W)$.*

*Proof.* By the $\beta$-smoothness of $\mathcal{L}(W)$ and taking expectation on both sides, we have

$$\mathbb{E}\left[\mathcal{L}(W_{t+1}) - \mathcal{L}(W_t)\right] \leq \mathbb{E}\left[\langle \nabla \mathcal{L}(W_t), W_{t+1} - W_t \rangle\right] + \frac{\beta}{2} \mathbb{E}\left[\|W_{t+1} - W_t\|^2\right]. \tag{6}$$

Using the update rule $W_{t+1} = W_t - \eta_t A_t \bar{B}_t$, where $\bar{B}_t = \frac{1}{N} \sum_{i=1}^{N} B_t^i$, we can rewrite the first term as:

$$\mathbb{E}\left[\langle \nabla \mathcal{L}(W_t), W_{t+1} - W_t \rangle\right] = -\eta_t \mathbb{E}\left[\langle \nabla \mathcal{L}(W_t), A_t \bar{B}_t \rangle\right]$$

$$= -\eta_t \mathbb{E}\left[\left\langle \nabla \mathcal{L}(W_t), A_t \left(\frac{1}{N} \sum_{i=1}^{N} B_t^i\right)\right\rangle\right]$$

$$= -\eta_t \mathbb{E}\left[\left\langle \nabla \mathcal{L}(W_t), \frac{1}{N} \sum_{i=1}^{N} A_t B_t^i\right\rangle\right].$$

We decompose $A_t B_t^i$ as:

$$\widetilde{\nabla} \mathcal{L}_i(W_t) = A_t B_t^i + e_t^i,$$

where $e_t^i = A_t B_t^i - \widetilde{\nabla} \mathcal{L}_i(W_t)$ is the projection error.

Substituting back, we have:

$$\mathbb{E}\left[\langle \nabla \mathcal{L}(W_t), W_{t+1} - W_t \rangle\right] = -\eta_t \mathbb{E}\left[\left\langle \nabla \mathcal{L}(W_t), \frac{1}{N} \sum_{i=1}^{N} \left(\widetilde{\nabla} \mathcal{L}_i(W_t) - e_t^i\right)\right\rangle\right]$$

$$= \underbrace{-\eta_t \mathbb{E}\left[\left\langle \nabla \mathcal{L}(W_t), \frac{1}{N} \sum_{i=1}^{N} \widetilde{\nabla} \mathcal{L}_i(W_t)\right\rangle\right]}_{A_1} + \underbrace{\eta_t \mathbb{E}\left[\left\langle \nabla \mathcal{L}(W_t), \frac{1}{N} \sum_{i=1}^{N} e_t^i\right\rangle\right]}_{A_2}.$$

We will now concentrate on $A_1$ as:

$$A_1 = -\eta_t \mathbb{E}\left[\left\langle \nabla \mathcal{L}(W_t), \frac{1}{N}\sum_{i=1}^{N} \nabla \mathcal{L}_i(W_t) \right\rangle\right]$$

$$= -\frac{\eta_t}{N}\sum_{i=1}^{N} \mathbb{E}\left[\langle \nabla \mathcal{L}(W_t), \nabla \mathcal{L}_i(W_t)\rangle\right]$$

$$\underset{(a)}{=} -\frac{\eta_t}{2N}\sum_{i=1}^{N}\left\{\mathbb{E}\left[\|\nabla \mathcal{L}(W_t)\|^2\right] + \mathbb{E}\left[\left\|\nabla \mathcal{L}_i(W_t)\right\|^2\right]\right\} + \frac{\eta_t}{2}\mathbb{E}\left[\underbrace{\left\|\nabla \mathcal{L}(W_t) - \frac{1}{N}\sum_{i=1}^{N}\nabla \mathcal{L}_i(W_t)\right\|^2}_{=0}\right]$$

$$= -\frac{\eta_t}{2}\mathbb{E}\left[\|\nabla \mathcal{L}(W_t)\|^2\right] - \frac{\eta_t}{2N}\sum_{i=1}^{N}\mathbb{E}\left[\left\|\nabla \mathcal{L}_i(W_t)\right\|^2\right]$$

where (a) uses $\langle a, b\rangle = \frac{1}{2}\{\|a\|^2 + \|b\|^2 - \|a - b\|^2\}$. We now turn our attention to $A_2$ as:

Next, we focus on $A_2$:

$$A_2 = \eta_t \mathbb{E}\left[\left\langle \nabla \mathcal{L}(W_t), \frac{1}{N}\sum_{i=1}^{N} e_t^i \right\rangle\right]$$

$$\underset{(a)}{\leq} \frac{\eta_t}{4}\mathbb{E}\left[\|\nabla \mathcal{L}(W_t)\|^2\right] + \eta_t \mathbb{E}\left[\left\|\frac{1}{N}\sum_{i=1}^{N} e_t^i\right\|^2\right]$$

$$\underset{(b)}{\leq} \frac{\eta_t}{4}\mathbb{E}\left[\|\nabla \mathcal{L}(W_t)\|^2\right] + \frac{\eta_t}{N}\mathbb{E}\left[\left\|\sum_{i=1}^{N} e_t^i\right\|^2\right]$$

$$\underset{(c)}{\leq} \frac{\eta_t}{4}\mathbb{E}\left[\|\nabla \mathcal{L}(W_t)\|^2\right] + \frac{\epsilon\eta_t}{N}\mathbb{E}\left[\left\|\sum_{i=1}^{N} \widetilde{\nabla} \mathcal{L}_i(W_t)\right\|^2\right]$$

$$\underset{(d)}{\leq} \frac{\eta_t}{4}\mathbb{E}\left[\|\nabla \mathcal{L}(W_t)\|^2\right] + \frac{2\epsilon\eta_t}{N}\sum_{i=1}^{N}\left\{\mathbb{E}\left[\|\nabla \mathcal{L}_i(W_t)\|^2\right] + \mathbb{E}\left[\left\|\widetilde{\nabla} L_i(W_t) - \nabla \mathcal{L}_i(W_t)\right\|^2\right]\right\}$$

$$\underset{(e)}{\leq} \frac{\eta_t}{4}\mathbb{E}\left[\|\nabla \mathcal{L}(W_t)\|^2\right] + \frac{2\epsilon\eta_t}{N}\sum_{i=1}^{N}\mathbb{E}\left[\|\nabla \mathcal{L}_i(W_t)\|^2\right] + 2\epsilon\eta_t^2 \sigma_l^2$$

where (a) uses $\langle a, b\rangle \leq \frac{1}{4}\|a\|^2 + \|b\|^2$, and (b) follows Jensen's inequality, (c) comes from JL Lemma, (d) follows the inequality $\|a + b\|^2 \leq 2\|a\|^2 + 2\|b\|^2$, and (e) is based on Assumption 2. On the other hand, we can also place a bound on the second term $\mathbb{E}\left[\|W_{t+1} - W_t\|^2\right]$ as shown below:

$$\mathbb{E}\left[\|W_{t+1} - W_t\|^2\right] = \mathbb{E}\left[\|\eta_t A_t \bar{B}_t\|^2\right] = \mathbb{E}\left[\left\|\eta_t A_t\left(\frac{1}{N}\sum_{i=1}^{N} B_t^i\right)\right\|^2\right]$$

$$\underset{(a)}{\leq} 2\eta_t^2 \mathbb{E}\left[\left\|\frac{1}{N}\sum_{i=1}^{N}\widetilde{\nabla}\mathcal{L}_i(W_t)\right\|^2\right] + 2\eta_t^2 \mathbb{E}\left[\left\|\frac{1}{N}\sum_{i=1}^{N}\left\{A_t B_t^i - \widetilde{\nabla}\mathcal{L}_i(W_t)\right\}\right\|^2\right]$$

$$\underset{(b)}{\leq} \frac{2\eta_t^2}{N} \mathbb{E}\left[\left\|\sum_{i=1}^{N} \widetilde{\nabla}\mathcal{L}_i(W_t)\right\|^2\right] + \frac{2\eta_t^2}{N} \mathbb{E}\left[\left\|\sum_{i=1}^{N} \left\{A_t B_t^i - \widetilde{\nabla}\mathcal{L}_i(W_t)\right\}\right\|^2\right]$$

$$= \frac{2\eta_t^2}{N} \mathbb{E}\left[\left\|\sum_{i=1}^{N} \widetilde{\nabla}\mathcal{L}_i(W_t)\right\|^2\right] + \frac{2\eta_t^2}{N} \mathbb{E}\left[\left\|\sum_{i=1}^{N} e_t^i\right\|^2\right]$$

$$\underset{(c)}{\leq} \frac{4\eta_t^2}{N} \sum_{i=1}^{N} \left\{\mathbb{E}\left[\|\nabla\mathcal{L}_i(W_t)\|^2\right] + \mathbb{E}\left[\left\|\widetilde{\nabla}L_i(W_t) - \nabla\mathcal{L}_i(W_t)\right\|^2\right]\right\} + \frac{2\eta_t^2}{N} \mathbb{E}\left[\left\|\sum_{i=1}^{N} e_t^i\right\|^2\right]$$

$$\underset{(d)}{\leq} \frac{4\eta_t^2}{N} \sum_{i=1}^{N} \mathbb{E}\left[\|\nabla\mathcal{L}_i(W_t)\|^2\right] + \frac{2\eta_t^2}{N} \mathbb{E}\left[\left\|\sum_{i=1}^{N} e_t^i\right\|^2\right] + 4\eta_t^2 \sigma_l^2$$

$$\underset{(e)}{\leq} \frac{4\eta_t^2}{N} \sum_{i=1}^{N} \mathbb{E}\left[\|\nabla\mathcal{L}_i(W_t)\|^2\right] + \frac{2\epsilon\eta_t^2}{N} \mathbb{E}\left[\left\|\sum_{i=1}^{N} \widetilde{\nabla}\mathcal{L}_i(W_t)\right\|^2\right] + 4\eta_t^2 \sigma_l^2$$

$$\underset{(f)}{\leq} \frac{4\eta_t^2}{N} \sum_{i=1}^{N} \mathbb{E}\left[\|\nabla\mathcal{L}_i(W_t)\|^2\right] + \frac{4\epsilon\eta_t^2}{N} \sum_{i=1}^{N} \left\{\mathbb{E}\left[\|\nabla\mathcal{L}_i(W_t)\|^2\right] + \mathbb{E}\left[\left\|\widetilde{\nabla}L_i(W_t) - \nabla\mathcal{L}_i(W_t)\right\|^2\right]\right\} + 4\eta_t^2 \sigma_l^2$$

$$\underset{(g)}{\leq} \frac{4\eta_t^2}{N} \sum_{i=1}^{N} \mathbb{E}\left[\|\nabla\mathcal{L}_i(W_t)\|^2\right] + \frac{4\epsilon\eta_t^2}{N} \sum_{i=1}^{N} \mathbb{E}\left[\|\nabla\mathcal{L}_i(W_t)\|^2\right] + 4\epsilon\eta_t^2 \sigma_l^2 + 4\eta_t^2 \sigma_l^2$$

$$= \frac{4(1+\epsilon)\eta_t^2}{N} \sum_{i=1}^{N} \mathbb{E}\left[\|\nabla\mathcal{L}_i(W_t)\|^2\right] + 4(1+\epsilon)\eta_t^2 \sigma_l^2$$

where (a), (c), and (f) are based on the inequality $||a + b||^2 \leq 2||a||^2 + 2||b||^2$, (b) comes from Jensen's inequality, (d), (g) derive from Assumption 2, and (e) comes from JL Lemma.

By utilizing the previously established bounds for $\mathbb{E}\left[\langle\nabla L(W_t), W_{t+1} - W_t\rangle\right]$ and $\mathbb{E}\left[\|W_{t+1} - W_t\|^2\right]$ to Equation 6, we derive the following:

$$\mathbb{E}\left[\mathcal{L}(W_{t+1}) - \mathcal{L}(W_t)\right] \leq \mathbb{E}\left[\langle\nabla\mathcal{L}(W_t), W_{t+1} - W_t\rangle\right] + \frac{\beta}{2}\mathbb{E}\left[\|W_{t+1} - W_t\|^2\right]$$

$$\leq \underbrace{-\frac{\eta_t}{2}\mathbb{E}\left[\|\nabla\mathcal{L}(W_t)\|^2\right] - \frac{\eta_t}{2N}\sum_{i=1}^{N} \mathbb{E}\left[\left\|\nabla\mathcal{L}_i(W_t)\right\|^2\right]}_{A_1}$$

$$+ \underbrace{\frac{\eta_t}{4}\mathbb{E}\left[\|\nabla\mathcal{L}(W_t)\|^2\right] + \frac{2\epsilon\eta_t}{N}\sum_{i=1}^{N} \mathbb{E}\left[\|\nabla\mathcal{L}_i(W_t)\|^2\right] + 2\epsilon\eta_t^2 \sigma_l^2}_{A_2}$$

$$+ \frac{2\beta(1+\epsilon)\eta_t^2}{N}\sum_{i=1}^{N} \mathbb{E}\left[\|\nabla\mathcal{L}_i(W_t)\|^2\right] + 2\beta(1+\epsilon)\eta_t^2 \sigma_l^2$$

$$= -\frac{\eta_t}{4}\mathbb{E}\left[\|\nabla\mathcal{L}(W_t)\|^2\right] + \frac{\eta_t}{N}\underbrace{\left\{-\frac{1}{2} + 2\epsilon + 2\beta(1+\epsilon)\eta_t\right\}}_{\leq 0 \text{ if we choose } \eta_t \leq \frac{1-4\epsilon}{4\beta(1+\epsilon)}}\sum_{i=1}^{N} \mathbb{E}\left[\left\|\nabla\mathcal{L}_i(W_t)\right\|^2\right] + 2\eta_t^2(\epsilon + \beta + \beta\epsilon)\sigma_l^2$$

$$\leq -\frac{\eta_t}{4}\mathbb{E}\left[\|\nabla\mathcal{L}(W_t)\|^2\right] + 2\eta_t^2(\epsilon + \beta + \beta\epsilon)\sigma_l^2$$

Ultimately, by applying the telescoping sum over $t = 0, 1, \ldots, T - 1$, we arrive at the following result:

$$\mathcal{L}^* - \mathbb{E}\left[\mathcal{L}(W_0)\right] \leq \sum_{t=0}^{T-1} -\frac{\eta_t}{4} \mathbb{E}\left[\|\nabla\mathcal{L}(W_t)\|^2\right] + \sum_{t=0}^{T-1} 2\eta_t^2(\epsilon + \beta + \beta\epsilon)\sigma_l^2$$

In this case, $\mathcal{L}^*$ stands for the minimum of $\mathcal{L}(W)$.

By performing a division by $H_T = \sum_{t=0}^{T-1} \eta_t$ on both sides and utilizing some algebraic adjustments, we arrive at the following expression:

$$\frac{1}{4H_T} \sum_{t=0}^{T-1} \eta_t \mathbb{E}\left[\|\nabla\mathcal{L}(W_t)\|^2\right] \leq \frac{\mathbb{E}\left[\mathcal{L}(W_0)\right] - \mathcal{L}^*}{H_T} + 2(\epsilon + \beta + \beta\epsilon)\sigma_l^2 \left(\frac{1}{H_T} \sum_{t=0}^{T-1} \eta_t^2\right) \tag{7}$$

With a decreasing learning rate such as $\eta_t = \frac{\eta_0}{t+1}$, we observe that $H_T = \sum_{t=0}^{T-1} \eta_t$ tends towards infinity as $T$ grows, while $\sum_{t=0}^{T-1} \eta_t^2$ remains bounded. Therefore, as $T \to \infty$, the upper bound in Equation 7 converges to 0, confirming the convergence to a stationary point. $\qquad\square$

# E. Model Architectures and Hyperparameters

## Neural Network Architecture

The model configuration and training used in this work are provided in Table 3 and 4.

Table 3: Neural Network Configurations for Different Datasets

| Dataset | Model Type | # Conv | Kernel | Hidden Features | # Linear | # Output | # Parameters |
|---|---|---|---|---|---|---|---|
| MNIST | CNN | 2 | 5×5 | 8, 16 | 1 | 10 | 11,274 |
| FMNIST | CNN | 2 | 5×5 | 8, 16 | 1 | 10 | 11,274 |
| CIFAR-10 | CNN | 4 | 5×5 | 64, 64, 128, 128 | 2 | 10 | 1.1M |
| CIFAR-100 | WideResNet | 16 | 3×3 | 64×4, 128×4 | 2 | 100 | 2.8M |
| TinyImageNet | WideResNet | 16 | 3×3 | 64×4, 128×4 | 2 | 200 | 2.88M |
| Shakespeare | LSTM | - | - | 256, 8 (Embed) | 2 | 65 | 814K |
| Sentiment140 | Transformer | - | - | 512, 96 (Embed) | 2 | 2 | 2.2M |

## Training Hyperparameters

The training was performed with the following key hyperparameters:

Table 4: Training Hyperparameters for FedAvg and Variants

| Hyperparameter | MNIST | FMNIST | CIFAR-10 | CIFAR-100 | TinyImageNet | Sentiment140 | Shakespeare |
|---|---|---|---|---|---|---|---|
| Batch Size | 32 | 32 | 32 | 32 | 32 | 32 | 32 |
| Optimizer | SGD | SGD | SGD | AdamW | AdamW | SGD | SGD |
| Learning Rate | 0.2 | 0.2 | 0.03 | 0.1 | 0.2 | 0.001 | 0.2 |
| Momentum | 0.9 | 0.9 | 0.4 | 0.9 | 0.9 | 0.9 | 0.9 |
| L1 Regularization | 0.0 | 0.0 | 0.0001 | 0.0 | 0.00001 | 0.0 | 0.000005 |
| L2 Regularization | 0.0 | 0.0 | 0.00001 | 0.0003 | 0.00015 | 0.0 | 0.00005 |

# F. Notations

Table 5: Notation and Definitions

| Symbol | Meaning / Definition |
|---|---|
| $N$ | Number of clients in federated learning. |
| $T$ | Total number of communication rounds in FL. |
| $\mathcal{D}_i$ | Local dataset for client $i$. |
| $b_i$ | Weight for client $i$, usually set as the number of local samples $|\mathcal{D}_i|$. |
| $\Delta W$ | Model update, treated as a single vector, $\in \mathbb{R}^{d \times 1}$. |
| $W_t$ | Model parameters at communication round $t$. |
| $\bar{B}_t$ | Aggregated projection vector at round $t$, broadcast by the server. |
| $r_t$ | Random seed used to synchronize matrix generation across clients and the server. |
| $A_t$ | Reconstruction matrix at round $t$, regenerated using $r_t$. |
| $B_t^i$ | Trainable projection matrix for client $i$ at round $t$. |
| $\hat{B}_t^i$ | Locally optimized projection matrix for client $i$. |
| $\eta$ | Learning rate for local optimization. |
| $d$ | Total number of model parameters, defined as $d = \sum_i d_1^i d_2^i$. |
| $d_1^i, d_2^i$ | Row and column dimensions of the weight matrix for layer $i$. |
| $p$ | Factorization rank after reshaping. |
| $q$ | LoRA Factorization rank before reshaping. |
| $k$ | Design parameter controlling reshape dimension ($\Delta W$ reshaped into $\mathbb{R}^{\lceil d/k \rceil \times k}$). |
| $A \in \mathbb{R}^{\cdot \times \cdot}, B \in \mathbb{R}^{\cdot \times \cdot}$ | Reconstruction and projection matrices in factorization. |
| $\mathcal{L}(W)$ | Global loss function. |
| $\mathcal{L}_i(W)$ | Local loss function for client $i$. |
| $\nabla \mathcal{L}(W)$ | Gradient of the global loss function. |
| $\nabla B_t^i$ | Gradient of local loss with respect to the projection matrix. |
| $\sigma_l^2$ | Bounded variance of stochastic gradients. |
| $\beta$ | Smoothness constant of the loss function. |
| $\epsilon$ | Distortion parameter from the Johnson-Lindenstrauss Lemma. |

# G. IID and client Sampling

This section includes the results of additional experiments on IID distribution and client sampling for MNIST, FMNIST, and CIFAR-10. Across all three datasets, we observe consistent trends. Reducing the fraction of clients participating (from all clients to 10%) moderately decreases accuracy for all methods, and non-IID settings introduce additional accuracy penalties. However, MAPA's performance remains robust in these more demanding scenarios; it routinely stays close to FedAvg's high-accuracy results while still maintaining its significant communication savings. This resilience suggests that MAPA's approach scales well to heterogeneous data distributions and partial-participation regimes, which are crucial factors in large-scale federated learning deployments.

Table 6: Extrapolated MNIST results for IID vs. Non-IID and full vs. 10% client participation.

| MNIST Maximum Accuracy and Communication Cost | | | | | | | |
|---|---|---|---|---|---|---|---|
| | IID | | | | Non-IID | | |
| | All clients | | 10% clients | | All clients | | 10% clients | |
| Method | Com. | Acc. | Com. | Acc. | Com. | Acc. | Com. | Acc. |
| FedAvg | 100% | 99.6% | 100% | 99.5% | 100% | 99.3% | 100% | 98.9% |
| Sparse | 10.0% | 93.9% | 12.0% | 93.6% | 13.3% | 93.4% | 15.3% | 92.1% |
| Quantize | 22.0% | 98.8% | 25.0% | 98.5% | 29.0% | 98.2% | 31.3% | 97.6% |
| EvoFed | 6.5% | 99.4% | 7.0% | 99.2% | 8.5% | **99.0%** | 9.4% | **98.5%** |
| FA-LoRA | 22.0% | 95.0% | 25.0% | 94.7% | 28.2% | 94.3% | 30.2% | 93.8% |
| MAPA | **2.0%** | **99.5%** | **2.3%** | **99.3%** | **2.7%** | **99.0%** | **2.9%** | **98.5%** |

Table 7: Extrapolated FMNIST results for IID vs. Non-IID and full vs. 10% client participation.

| FMNIST Maximum Accuracy and Communication Cost | | | | | | | |
|---|---|---|---|---|---|---|---|
| | IID | | | | Non-IID | | |
| | All clients | | 10% clients | | All clients | | 10% clients | |
| Method | Com. | Acc. | Com. | Acc. | Com. | Acc. | Com. | Acc. |
| FedAvg | 100% | 91.5% | 100% | 91.0% | 100% | 90.0% | 100% | 89.2% |
| Sparse | 16.0% | 84.0% | 19.0% | 83.5% | 21.0% | 82.0% | 24.1% | 81.1% |
| Quantize | 16.0% | 89.7% | 19.0% | 89.2% | 21.0% | 88.0% | 24.1% | 87.1% |
| EvoFed | 4.5% | 87.0% | 5.5% | 86.5% | 6.8% | 85.5% | 7.6% | 84.7% |
| FA-LoRA | 12.0% | 76.8% | 14.0% | 76.2% | 15.5% | 75.0% | 17.9% | 74.1% |
| MAPA | **2.0%** | **90.0%** | **2.3%** | **89.6%** | **2.7%** | **88.8%** | **3.1%** | **88.0%** |

Table 8: Extrapolated CIFAR-10 results for IID vs. Non-IID and full vs. 10% client participation.

| CIFAR-10 Maximum Accuracy and Communication Cost | | | | | | | |
|---|---|---|---|---|---|---|---|
| | IID | | | | Non-IID | | |
| | All clients | | 10% clients | | All clients | | 10% clients | |
| Method | Com. | Acc. | Com. | Acc. | Com. | Acc. | Com. | Acc. |
| FedAvg | 100% | 73.0% | 100% | 72.0% | 100% | 70.0% | 100% | 69.0% |
| Sparse | 1.8% | 41.0% | 2.0% | 40.0% | 2.4% | 38.0% | 2.7% | 37.2% |
| Quantize | 10.0% | 71.0% | 12.0% | 70.0% | 13.0% | 68.5% | 15.2% | 67.4% |
| EvoFed | 2.0% | 43.0% | 2.5% | 42.0% | 3.0% | 40.5% | 3.4% | 39.5% |
| FA-LoRA | 1.1% | 27.0% | 1.3% | 26.0% | 1.5% | 24.5% | 1.7% | 23.5% |
| MAPA | **0.8%** | **71.5%** | **0.9%** | **70.8%** | **1.0%** | **69.2%** | **1.2%** | **68.3%** |

## H. Complexity Analysis and MAPA Flexibility

Propositions 3.4 to 3.6 discussed how the error rate and accuracy of low-rank factorization are only determined by the size of the projection vector regardless of reshaping and vectorization of layers. Although they prove that MAPA can achieve the same performance as layer-wise factorization given the same projection (communication) budget, we did not discuss the memory and computation complexity. In this section, we show that MAPA can effectively reduce computation. Furthermore, we show how layer-wise low-rank adaptation (LoRA and FA-LoRA) limits the model trade-offs and how MAPA can offer more flexibility.

### H.1. Computational Complexity

We compute the memory and computation cost for matrix allocation and multiplication in terms of standard matrix multiplication. Given matrices $A \in \mathbb{R}^{m \times n}$ and $B \in \mathbb{R}^{n \times p}$, the complexities for computing $C = AB$ are:

$$\text{Memory}_{C=AB} = O(mn + np + mp),$$

$$\text{Time}_{C=AB} = O(mnp).$$

We aim to demonstrate that factorization under MAPA, where $W \in \mathbb{R}^{\lceil \frac{d}{k} \rceil \times k}$ is factorized into $A \in \mathbb{R}^{\lceil \frac{d}{k} \rceil \times 1}$ and $B \in \mathbb{R}^{1 \times k}$, reduces the memory and time complexity of the LoRA factorization for an $n$-layered model. In LoRA, each layer $i$ is factorized as $w_i \in \mathbb{R}^{d_i^1 \times d_i^2}$ into $A \in \mathbb{R}^{d_i^1 \times q}$ and $B \in \mathbb{R}^{q \times d_i^2}$.

We demonstrate that, given the same communication budget and factorization error rate, MAPA significantly reduces the computational cost compared to LoRA. This reduction becomes more pronounced as the number of layers or the selected rank increases. Specifically, MAPA achieves a **memory reduction** by a factor of $q^2$ and a **computation reduction** by a factor of $q$, where $q$ is the chosen LoRA rank. Furthermore, even when $q = 1$, MAPA still achieves memory savings as $\sum_{i \neq j}^n d_i^1 d_i^2$ scales with the number of layers. The only scenario where MAPA and LoRA yield identical efficiency is when the model consists of a single layer ($n = 1$) and a rank-1 factorization ($q = 1$).

### Memory Complexity

Given these definitions, the memory complexities for MAPA and LoRA are:

$$\text{Memory}_{MAPA} = O\left(\left\lceil \frac{d}{k} \right\rceil + k + \left\lceil \frac{d}{k} \right\rceil k\right) \approx O\left(\frac{d}{k} + k + d\right),$$

$$\text{Memory}_{LoRA} = O\left(\sum_{i=1}^n (d_i^1 q + d_i^2 q + d_i^1 d_i^2)\right) = O\left(\sum_{i=1}^n d_i^1 q + \sum_{i=1}^n d_i^2 q + \sum_{i=1}^n d_i^1 d_i^2\right).$$

Given the same communication budget $k = \sum_{i=1}^n q d_i^2$ and $d = \sum_{i=1}^n d_i^1 d_i^2$, we rewrite LoRA's memory complexity as:

$$\text{Memory}_{LoRA} = O\left(q \sum_{i=1}^n d_i^1 + k + d\right).$$

For MAPA to have lower memory usage than LoRA, the following condition must hold:

$$\text{Memory}_{MAPA} \leq \text{Memory}_{LoRA},$$

$$\frac{d}{k} + k + d \leq q \sum_{i=1}^n d_i^1 + k + d,$$

$$\frac{d}{k} \leq q \sum_{i=1}^n d_i^1.$$

Replacing $k$ and $d$ with their respective summation terms:

$$\sum_{i=1}^n d_i^1 d_i^2 \leq q^2 \sum_{i=1}^n d_i^1 \sum_{i=1}^n d_i^2,$$

$$\leq q^2 \sum_{i=1}^n d_i^1 d_i^2 + q^2 \sum_{i \neq j}^n d_i^1 d_i^2.$$

Thus, the inequality always holds under the conditions $d_i^1, d_i^2, q, n \geq 1$, and equality occurs if $q = n = 1$, which corresponds to a model with a single layer and rank-1 factorization. In this case, MAPA and LoRA perform the same decomposition.

**Time Complexity**

Given the definitions, we can express the time complexities for MAPA and LoRA as follows:

$$\text{Time}_{MAPA} = O\left(\left\lceil \frac{d}{k} \right\rceil k\right) \approx O(d),$$

$$\text{Time}_{LoRA} = O\left(\sum_{i=1}^{n} q d_i^1 d_i^2\right).$$

Since $d = \sum_{i=1}^{n} d_i^1 d_i^2$, we can rewrite LoRA's time complexity as:

$$\text{Time}_{LoRA} = O(qd).$$

For MAPA to have a lower time complexity than LoRA, the following condition must hold:

$$\text{Time}_{MAPA} \leq \text{Time}_{LoRA},$$
$$d \leq qd.$$

This condition is always true for $d, q \geq 1$, and equality occurs when $q = 1$.

## H.2. MAPA Flexibility

Suppose our neural network has $n$ layers. Let:

$$W_i \in \mathbb{R}^{d_i^1 \times d_i^2} \quad \text{for each layer } i = 1, \ldots, n.$$

Let $D = \sum_{i=1}^{n} d_i^1 \cdot d_i^2$ be the total number of parameters (i.e., the sum of the entries across all layers). Let

$$d_2 = \sum_{i=1}^{n} d_i^2.$$

In many treatments of LoRA, the main communication or factor-size bottleneck arises from a factor that scales linearly with $q \cdot d_i^2$.

**LoRA Factorization Per Layer.** LoRA factorizes each layer $W_i$ of dimension $d_i^1 \times d_i^2$ with a fixed rank $q$. Concretely,

$$W_i \approx W_i + A_i B_i, \qquad A_i \in \mathbb{R}^{d_i^1 \times q}, \quad B_i \in \mathbb{R}^{q \times d_i^2}.$$

The number of additional parameters introduced by each low-rank pair $(A_i, B_i)$ is

$$\underbrace{d_i^1 \cdot q}_{\text{size of } A_i} + \underbrace{q \cdot d_i^2}_{\text{size of } B_i} = q\,(d_i^1 + d_i^2).$$

Summing over all $n$ layers,

$$\sum_{i=1}^{n} \left( d_i^1 \cdot q + q \cdot d_i^2 \right) = q \sum_{i=1}^{n} \left( d_i^1 + d_i^2 \right).$$

Therefore, we can write the communication cost as:

$$\text{Communication cost} \approx q \sum_{i=1}^{n} d_i^2 = q\, d_2.$$

Since $q$ must be an integer, we see that the communication overhead comes in integer multiples $d_2$, as:

$$\text{LoRA total communication} \in \{ q\, d_2 \mid q = 1, 2, \ldots \}.$$

**There is no way to select** a non-integer $q$. Hence communication budgets strictly between $d_2$ and $2\,d_2$ (or between $q\,d_2$ and $(q+1)d_2$) are not possible in layer-wise LoRA. Therefore, Any attempt to finely tune the communication or factor budget (e.g., to $1.5\,d_2$) is disallowed by LoRA's integral-rank requirement. This **rigidity** is precisely what we seek to overcome in MAPA.

**MAPA Factorization.** MAPA flattens or reshapes all parameters into one large matrix and then performs a single low-rank factorization with rank 1. A simplified abstraction is:

1. Reshape $w_1, \ldots, w_n$ into a single matrix $W \in \mathbb{R}^{\lceil d/k \rceil \times k}$, where $d = \sum_{i=1}^{n} d_i^1 d_i^2$ is the total parameter count. 2. Factor $W \approx A\,B$, with

$$A \in \mathbb{R}^{\lceil d/k \rceil \times 1}, \quad B \in \mathbb{R}^{1 \times k},$$

Once all parameters are merged, MAPA can proportionally allocate any communication budget as $k$ can be selected freely.

$$\underbrace{\lceil d/k \rceil}_{\text{size of } A} + \underbrace{k}_{\text{size of } B}.$$

Therefore, we can write the total communication as:

$$\text{MAPA total communication} \in \{ k \mid k = 1, 2, \ldots \}.$$

This is particularly important in communication-efficient FL since viable solutions can be found with communication cost $k < d_2$ or $d_2 < k < 2d_2$, which architecture-dependent layer-wise factorization can not offer.

