# OpenReview forum: "Communication Efficient Federated Learning via Model-Agnostic Projection Adaptation"
_ICML.cc/2025/Conference — Submitted to ICML 2025_

### Official Review · Reviewer_JjTg · 2025-03-03

**Overall Recommendation:** 3

**Summary:**

The authors propose a new method called MAPA for parameter-efficient federated fine-tuning. The main advantages of MAPA are that it does not depend on the architecture, unlike other LoRA-based methods, and it reduces the computational and memory costs while providing a better or comparable performance. The authors provide a convergence guarantee of the proposed method under common assumptions with a decaying learning rate. The experimental results show the superiority of the proposed method.

**Claims And Evidence:**

I think the authors may want to clarify their claim about being not architecture-independent. I couldn't fully follow it. First, I thought their method was applicable to all types of neural net layers (CNN, etc.), unlike LoRA, which is only for linear layers. If this is the case (I can see that you use some CNN architectures in the experiment), I wonder how it is happening.

**Essential References Not Discussed:**

I think one missing part in the intuition of the integration of the single reshaped matrix update/factorization idea is any other work using this in a centralized setting. If this has not been done before in a centralized setting, then it means that the authors' contribution may be an even wider setting, i.e., a centralized setting. If there exist similar ideas in centralized LoRA literature, mentioning them in the related works would be good. Please correct me if I am mistaken.

**Experimental Designs Or Analyses:**

The experiments sound good overall. I have a few questions, which can be found in the questions and weaknesses section.

**Methods And Evaluation Criteria:**

Yes, it looks good overall. My questions are in the weaknesses and questions section.

**Other Comments Or Suggestions:**

1. In Fig. 2, why are the methods compared in a centralized setting? To my knowledge, for example, FFA-LoRA ([SLLD'24] in ICLR24) also solves the exact aggregation problem (with updates $A_i$, $B_i$s from clients, how to aggregate them is a question), which is specific to the federated setting.
2. I think there is a typo in the proof where $e_t^i$ is defined. I guess it should be just the negation of the written expression.

**Other Strengths And Weaknesses:**

- Strengths:
1. The paper is written very clearly. I appreciate the authors' presentation.
2. The proposed method's intuition is well-explained.
3. The theoretical guarantee is a plus considering many works in the literature lack it.
4. The experiments show the superiority of the proposed method compared to the baselines in terms of communication efficiency and training quality.

- Weaknesses:

1. I don't fully understand what is wrong with layer-by-layer separation (let's say we select some fixed $k$ for all layers) in terms of architecture dependence. Cannot the proposed method (first reshaping the matrix and separating A and B) be applied layer by layer for any architecture type following a similar method? Do the authors have an ablation experiment where they apply their technique layer by layer to the models and compare with the current version?
2. The model backbone is updated at every round as in Eq. (1). This may create the following problem: What makes LoRA advantageous is that at the end of fine-tuning, we have some small number of parameters that can easily be merged with or separated from the original model. Here, if we update the model backbone and initialize different A matrices at every iteration, we won't be able to have a low rank representation of the fine-tuned part in the end. Yes, we can separate the fine-tuned part ($\sum_t A_t\bar{B}_t$), but it will take a memory size of $d$, unlike the low storage cost of LoRA parameters.
3. There are many newer techniques in Federated LoRA literature. I would expect a comparison with a few more recent and solid FL LoRA baselines.
4. I think the forward pass of the proposed method should be slower than the other LoRA methods. In the proposed one, $\Delta W_i$ s in every layer $i$ is a full dimensional matrix. However, in the LoRA versions, it is split into in order to have $2\times d\times q$ instead of large $d^2$ (Here, $d$ and $q$ represent the full and LoRA dimensions within a layer). Can authors elaborate on it?

**Questions For Authors:**

1. I have found the propositions and the same reconstruction error with a smaller number of parameters solid. However, I question if it is practical with first-order methods, i.e., SGD, etc.  What I mean is that I agree that the reconstruction error given in Defn 3.3 holds for the best A and B parameter set. However, it might be the case that we cannot find A and B with just gradient-based optimization (what actually happens in our setting). In that case, one might claim that having more parameters may find better A and B with a practically lower reconstruction error. Do authors have an explanation for this or an experimental result supporting it?
2. Shouldn't Theorem 4.3 be a high probability bound due to JL lemma? If so, I would suggest stating that the bound holds for a high probability of (the probability at least, e.g., $1-T\epsilon$..).
3. In experiments, how are these models pre-trained? Since we compare fine-tuning techniques, sharing how the models are actually pre-trained is important.

**Relation To Broader Scientific Literature:**

The proposed method seems to improve the federated fine-tuning in communication efficiency while providing better or comparable performance. Compared to the previous literature, I have found the explanations and theoretical analysis good. The experimental results are superior to the selected baselines.

**Theoretical Claims:**

I didn't follow the proofs of Propositions, but they seem to make sense. I skimmed through the proof of the convergence analysis. It seemed reasonable to me.

---

> ### Author Rebuttal · Authors · 2025-04-01
>
> ## 1. **Architecture-independency**
> You are correct. Our approach flattens the gradients and factorizes them in a matrix form rather than directly factorizing the parameters. This gradient-based factorization is independent of specific architectural details, making it applicable to any model.
> ## 2. **Centralized setting**
> We discussed this issue in **response 5 to reviewer NJUL** and presented the experimental results in **response 1 to reviewer FKCh**.
> ## 3. **Layer-wise factorization vs. Entire model factorization**
> While layer-wise factorization is possible, it introduces following practical issues:
>
> 1. **Memory Overhead**: For reducing model communication $k$-fold, global factorization on $W^d$ results in $W^{k * d/k} = A^{k * 1} B^{1 * d/k}$, while layer-wise factorization of an $n$-layer model, given layer-$i$'s parameters as $W_i^{d_i}$ will be $W_i^{k * d_i/k} = A_i^{k * 1} B_i^{1 * d_i/k}$  requires storing $n$ separate $A_i$, increasing memory overhead $n$ times.
>
> 2. **Architecture Constraints**: In global model factorization, we can essentially decide on any arbitrary rate of compression. In contrast, layer-wise factorization faces limitations, as choosing $k$ higher than the layer size is impossible.
> As shown experimentally in **response 3 of reviewer FKCh**, global factorization enables effective fine-tuning even at **10k-fold** compression for a model with **357M** parameters. Meanwhile, individual compression of a 1024-parameter layer at 10k-fold is not possible.
>
> 3. **Suboptimal Performance**: Lastly, global factorization outperforms layer-wise factorization since layer-wise allocates relatively equal expression budgets $d_i/k$ across layers, regardless of their gradient magnitudes. This leads to suboptimal communication budgeting, unlike global factorization, which allocates a higher expressivity budget to high magnitude gradients and higher compression to less informative gradients.
>
> Additional experiments on QNLI and SST2 illustrate this suboptimality:
> Model|SST2 Acc|Round@80%|QNLI Acc|Round@80%
> -|-|-|-|-
> Layered$_{1k}$|95.94|10|88.98|18
> MAPA$_{1k}$|96.41|7|92.58|15
> Layered$_{10k}$|92.27|12|82.42|29
> MAPA$_{10k}$|94.92|9|90.86|19
> ## 4. **Backbone is updated**
> You are correct. Unlike PEFT methods such as LoRA, MAPA does not inherently reduce the number of model parameters, as the full model backbone is updated at each communication round. Instead, MAPA primarily reduces gradient communication overhead in FL, as minimizing communication overhead is typically more critical than reducing parameter storage in this setting.
> ## 5.  **Comparison with more recent baselines**
> Many FL LoRA studies address broader challenges than communication. We used FA-LoRA as a central baseline due to its communication focus and conceptual similarity. Per your suggestion, we added LoRA and SA-LoRA to our baselines for LLM fine-tuning comparisons. (see **response 3 to reviewer FKCh**).
> ## 6. **Forward pass cost**
> LoRA adds computational overhead in the forward pass by including additional low-rank adaptation layers. The computation $y = Wx + BAx$ incurs complexity:
> - Frozen parameters: $O(d^2)$
> - LoRA layers: Two multiplications $Ax$ and $B(Ax)$: $O(2dq)$
>
> Thus, LoRA’s total forward pass complexity is $O(d^2 + 2dq)$.
> In contrast, MAPA applies low-rank factorization **only during the backward pass**, leaving the forward pass complexity unchanged at $O(d^2)$.
> ## 7. **Why is it centralized in Fig. 2?**
> In Fig. 2, we used a single-client setup to isolate the intrinsic performance impact of gradient compression, independent of data heterogeneity or client sampling in FL. Thus, any differences reflect compression effectiveness alone.
> ## 8. **Typo**
> Thank you; we fixed it.
> ## 9. **As B is not the optimal point, can having more parameters in A lead to better convergence?**
> In MAPA, updates follow \(W_t = W_{t-1} + AB\), where the \(A\) is **fixed** within each round. Therefore, gradient $B$ computation is just the gradient \(\nabla W\) projection linearly onto the subspace spanned by \(A\), and SGD can reliably solve linear projection to find a near-optimal $B$.
> ## 10.**Theorem 4.3**
> We thank the reviewer for this insightful point. The JL lemma provides a *probabilistic* guarantee of low-distortion embeddings. As such, to be entirely rigorous, our convergence statement can be made a *high-probability* result by union-bounding the (small) failure probability $\delta$ over the $T$ rounds:
> - At each round, distortion $\le\epsilon$ holds with probability $\ge 1-\delta$.
> - By union bound, for *all* $T$ rounds simultaneously, the probability is $\ge 1 - T\delta$.
> - Conditional on that event, the same inequalities apply, and the convergence proof remains identical.
> We fully agree with you and will strengthen the theorem statement via the presented arguments.
> ## 11. **Pre-trained Models**
> Original experiments were trained from scratch. For LLM fine-tuning, we used the HuggingFace `FacebookAI/roberta-large` checkpoint.

---

### Official Review · Reviewer_FKCh · 2025-03-11

**Overall Recommendation:** 1

**Summary:**

The paper proposes Model-Agnostic Projection Adaptation (MAPA), an approach to reduce communication overhead in FL. MAPA improves upon existing low-rank adaptation (LoRA) methods by factorizing the entire model parameter space as a single matrix, as opposed to decomposing layers independently. This model-agnostic approach allows for flexible balancing of communication and accuracy by randomly regenerating the reconstruction matrix (one of the two matrices) in each round.

**Claims And Evidence:**

I have issues agreeing with stated contributions. The authors state as the first contribution the idea of applying LoRA at the model level instead of layer by layer. The original LoRA paper casts the idea in this way where I don't see any difference. It is correct that in the implementation they apply it layer by layer, but these are experimental and implementational details.

The convergence analysis is also unclear if it is new. The convergence of LoRA has already been analyzed. The proposed algorithm introduces one of the two matrices to be random but I don't see why this requires quite different proofs.

The proposed algorithm is efficient for small networks but I have scalability doubts. The experiments have been conducted only on small datasets. I wonder what would happen for fine tuning of an LLM.

**Essential References Not Discussed:**

none

**Experimental Designs Or Analyses:**

I have checked the main body. The issue are discussed above.

**Methods And Evaluation Criteria:**

not quite. They should do it on fine tuning of LLMs. It is easy to create FL settings based on standard fine tuning data.

**Other Comments Or Suggestions:**

None

**Other Strengths And Weaknesses:**

Discussed above.

**Questions For Authors:**

1. Why not using and unbiased gradient approach?
2. Assess performance on fine tuning of LLMs.
3. What is really new in the algorithm?

**Relation To Broader Scientific Literature:**

I'm unclear about contribution statements. The work has major overlap with the LoRA paper (and the offsprings).

**Theoretical Claims:**

I'm unsure what is new in convergence results and proofs.
I wonder why don't standard SGD/FL technique apply. One possible argument would be the handling of random reconstruction matrices. But this can be viewed as stochasticity in gradient computation. It seems that the assumption of unbiased gradient would imply convergence. As a result, one needs to only show that the gradient estimators are unbiased. There is no reason to believe they are not and the proof should not be that hard.

---

> ### Author Rebuttal · Authors · 2025-04-01
>
> ## 1. **Key contributions**
>
> Thank you for raising this important point. To clarify, there are two fundamentally different strategies for leveraging low-rank structures in optimization:
>
> 1. **Low-Rank Parameterization**
> 2. **Low-Rank Gradient Projection**
>
> MAPA explicitly utilizes the latter strategy, whereas LoRA and its variants follow the first approach.
>
> **Why LoRA should be applied layer-wise?** Low-rank parameterization methods, like LoRA, inherently depend on **layer-wise decomposition**, as reparameterization must preserve input/output dimensions for each layer to maintain forward pass compatibility. LoRA decomposes each layer’s weight matrix $W$ individually as $h = (W+BA)x=Wx + BAx$. Treating the entire model parameters as a single matrix violates compatibility due to nonlinear activations and differing layer dimensions. LoRA paper acknowledges this constraint (Hu et al., **LoRA [Page 4, Section 4]**).
>
> Given a model-level LoRA factorization of $W^{I *O}$ into $A^{I * r}$ and $B^{r * O}$, where $I$ and $O$ are input and output dimensions of the model and $r$ is the factorization rank, the forward pass will be reduced to $y = BA(x)$, which does not express any nonlinearity of the network.
>
> In contrast, MAPA employs **gradient factorization** rather than **parameter factorization**. By applying low-rank constraints directly on the gradient instead of the parameters, MAPA reduces the gradient size while fully preserving model capacity. This approach is not constrained by layer-wise decomposition or model architecture since factorization occurs **after** computing gradients via standard forward/backward passes. Further discussion on gradient factorization literature appears in **response 5 to reviewer NJUL**.
>
> ## 2. **Theoretical contribution**
>
> Our convergence analysis extends standard federated SGD proofs [3,4] by incorporating a **random projection** (via the JL lemma) that introduces distortion $\epsilon$. This affects both descent direction and update variance. Unlike works that assume a fixed subspace or no gradient projection, we rigorously track how random, time-varying subspaces influence FL convergence. When $\epsilon=0$, MAPA becomes FedAvg. For $\epsilon>0$, we add a factor $(\epsilon + \beta + \epsilon\beta)$ but retain the same $\mathcal{O}(1/\sqrt{T})$ rate.
>
> We will revise the manuscript to emphasize this distinction in our convergence analysis.
>
> ## 3. **Fine-tuning of LLMs on larger datasets**
>
> Based on your feedback, we conducted fine-tuning experiments of **RoBERTa-large** on five large datasets of GLUE tasks. We evaluated MAPA, alongside *LoRA*, *FA-LoRA*, and *SA-LoRA* [2].
>
> The **1st Table** below compares the number of trainable parameters and communication load per round for each baseline.
>
> The **2nd Table** summarizes the results of fine-tuning, in which communication efficiency is evaluated by the number of rounds and the total communication needed to **reach 80%** accuracy, and the **3rd Table** presents the results for centralized LLM fine-tuning.
>
> The experiments used base code from [12], following the experimental setup and parameters from [2] for 300 FL rounds,
>
> References are located in **response 4 to reviewer Wq8k**.
>
> ---
> 1st Table:
> | Method | # Train Param | # Com. Param / Round |
> |-|-|-|
> | LoRA | 1.83M | 0.78M |
> | FFA-LoRA | 1.44M | 0.39M |
> | SA-LoRA | 1.83M | 0.39M |
> | MAPA$_{d/1k}$ | 357M | 0.36M |
> | MAPA$_{d/10k}$ | 357M | 35.70K |
> | MAPA$_{d/100k}$ | 357M | 3.57K |
> | MAPA$_{d/1m}$ | 357M | 357 |
>
> ---
>
> 2nd Table, FL fine-tuning:
> Model|SST2 Acc|SST2 Round|SST2 Total|QNLI Acc|QNLI Round|QNLI Total|RTE Acc|RTE Round|RTE Total|MNLIm Acc|MNLIm Round|MNLIm Total|MNLImm Acc|MNLImm Round|MNLImm Total
> -|-|-|-|-|-|-|-|-|-|-|-|-|-|-|-
> LoRA|84.86|36|28.08M|91.72|85|66.30M|86.62|180|140.40M|87.41|86|67.08M|87.34|82|63.96M
> FA-LoRA|94.15|44|17.16M|91.63|76|29.64M|57.28|—|—|85.92|76|29.64M|86.46|213|83.07M
> SA-LoRA|95.41|19|7.41M|91.04|55|21.45M|70.01|—|—|89.44|29|11.31M|85.49|126|49.14M
> MAPA$_{d/1k}$|96.79|5|1.78M|93.14|11|3.93M|87.91|23|8.21M|88.90|17|6.07M|88.26|22|7.85M
> MAPA$_{d/10k}$|96.10|5|178.50K|92.57|8|285.60K|89.57|23|821.10K|88.81|18|642.60K|87.43|25|892.50K
> MAPA$_{d/100k}$|95.53|5|17.85K|89.24|7|24.99K|84.38|24|85.68K|85.04|20|71.40K|84.60|29|103.53K
> MAPA$_{d/1m}$|90.37|7|2.50K|80.09|34|12.14K|57.04|—|—|72.46|—|—|37.76|—|—
>
> ---
>
> 3rd Table, centralized:
> Model|SST2 Acc|SST2 Round|SST2 Total|QNLI Acc|QNLI Round|QNLI Total|MNLI Acc|MNLI Round|MNLI Total
> -|-|-|-|-|-|-|-|-|-
> LoRA|95.23|51|39.78M|88.20|111|86.58M|85.23|132|102.96M
> FFA-LoRA|87.50|48|18.72M|68.05|—|—|86.48|66|25.74M
> SA-LoRA|94.69|110|42.90M|88.20|111|43.29M|86.02|62|24.18M
> MAPA$_{d/1k}$|95.47|9|3.21M|92.58|15|5.36M|86.80|37|13.21M
> MAPA$_{d/10k}$|94.61|8|0.28M|90.86|19|0.68M|85.00|38|1.36M
> MAPA$_{d/100k}$|79.38|—|—|83.83|18|64.26K|75.47|—|—
> MAPA$_{d/1m}$|58.52|—|—|56.56|—|—|37.81|—|—
>
> Overall, it can be seen that MAPA has the potential to enhance fine-tuning performance in centralized training too.

---

> > ### Comment · Reviewer_FKCh · 2025-04-01
> >
> > Thanks for providing the answers. I have no further questions and comments.

---

> > > ### Author Response · Authors · 2025-04-02
> > >
> > > Thank you very much for acknowledging our response, and we are pleased that all your concerns have been addressed.
> > >
> > > We would greatly appreciate it if you could update your score accordingly.
> > >
> > > Best regards, Authors of Paper 2961
> > >
> > > -----------------------------
> > > Edit:
> > >
> > > Dear Reviewer FKCh,
> > >
> > > We thank you so much for taking the time to review our paper and for your helpful suggestions.
> > >
> > > The author-reviewer discussion period ends soon. With the time ticking, we are getting very anxious.
> > > We did our best to provide answers to the questions and concerns you raised, including conducting extra experiments.
> > > You indicated you have no further questions and comments. We thank you for your prompt response.
> > >
> > > May we respectfully request that you reevaluate your score, unless you have further issues?
> > > We would be grateful.
> > >
> > > Best Wishes - authors

---

### Official Review · Reviewer_Wq8k · 2025-03-13

**Overall Recommendation:** 3

**Summary:**

This paper aims to improve communication efficiency in federated learning by proposing a new parameter factorization method. The proposed method is evaluated on seven public datasets and shows improved performance.

**Claims And Evidence:**

The claims are supported by method design and experimental validations.

**Essential References Not Discussed:**

A work with similiar idea needs to be discussed.

Jeong, Wonyong, and Sung Ju Hwang. "Factorized-fl: Personalized federated learning with parameter factorization & similarity matching." Advances in Neural Information Processing Systems 35 (2022): 35684-35695.

**Experimental Designs Or Analyses:**

The experimental design and analysis are sound in general.

**Methods And Evaluation Criteria:**

The proposed method and evaluation make sense in general but lack some comparison.

**Other Comments Or Suggestions:**

NA

**Other Strengths And Weaknesses:**

**Strength**
- Improving communication is an important topic in federated learning.

- The motivation for improving the LORA-based method is well demonstrated.

- The proposed method shows improvements in both communication and performance.


**Weakness**

- The proposed method approximates the updates of all layers by adjusting matrix B only, which may harm the model’s ability to explore richer subspaces.

- The design of single vector factorization shares a similar idea from [1], which needs to be included in the discussion and experiment comparison.

- It is not clear how good the convergence bound is compared with the FedAvg convergence, and in addition, what is the practical implication of this convergence analysis.


[1] Jeong, Wonyong, and Sung Ju Hwang. "Factorized-fl: Personalized federated learning with parameter factorization & similarity matching." Advances in Neural Information Processing Systems 35 (2022): 35684-35695.

**Questions For Authors:**

Please see the weakness part.

**Relation To Broader Scientific Literature:**

This paper contributes to the general federated learning community.

**Theoretical Claims:**

The theoretical claims and proofs look correct.

---

> ### Author Rebuttal · Authors · 2025-04-01
>
> ## 1. **Only updating B**
>
> Thank you for highlighting this concern. Indeed, relying solely on $B$ limits subspace exploration, as seen in FA-LoRA’s performance decline, SA-LoRA [2], and Figure 7. MAPA addresses this by **randomizing $A$ each round**, promoting diverse subspaces. Figure 7 shows that fixing $A$ at low ranks severely degrades accuracy, whereas randomizing $A$ maintains performance. We further verified this advantage in **response 2, reviewer NJUL**.
>
>
> ## **2. Comparison with [1]**
>
> We appreciate your mentioning Factorized-FL. Below are key distinctions alongside comparative experiments under the same setup:
>
> The key difference is that Factorized-FL applies factorization on layer-wise parameters, whereas MAPA factorizes the gradient of the entire model. In response to similar questions, we previously elaborated on why gradient (**response 1, reviewer FKCh**)  and model-level factorization (**response 3, reviewer JjTg**) can outperform layer-wise parameter factorization.
>
> Although both methods use rank-1 factorization, in Factorized-FL, rank-1 is a hyperparameter needing tuning or increasing for larger models to avoid limiting representation capacity. In MAPA, rank-1 is inherent and does not restrict model capacity; instead, the reshaping factor $k$ determines the compression rate. Consequently, Factorized-FL's communication per round is constrained by model dimensions, while MAPA can compress gradients to arbitrary degrees independent of architecture dimensions.
>
> Factorized-FL is similar to a rank-1 LoRA architecture, with a sparse bias matrix replacing LoRA’s frozen fine-tuned parameter, initialized as zero. LoRA imposes strict regularization to preserve pre-trained parameters, whereas Factorized-FL employs softer regularization, allowing updates when necessary.
>
> Factorized-FL emphasizes personalized FL by sharing one vector globally and keeping the other client-specific. To directly compare factorization effectiveness with MAPA, one could share both vectors globally. However, as noted by the authors [1] (**Page 6, Personalized Weight Averaging**), sharing both vectors significantly increases communication load, adversely affecting efficiency.
> Below, we highlight this fact by comparing **global model** training on CIFAR-10 and SVHN under IID and non-IID splits. **“Com@X%”** indicates total communication needed to reach X% of FedAvg’s final accuracy:
>
> |Method|CIFAR10 Com@80%|CIFAR10 Com@90%|CIFAR10-N Com@80%|CIFAR10-N Com@90%|SVHN Com@80%|SVHN Com@90%|SVHN-N Com@80%|SVHN-N Com@90%|Com/Round|
> |-|-|-|-|-|-|-|-|-|-|
> |FedAvg|305.85|407.80|326.24|652.48|183.51|244.68|285.46|509.75|20.39GB|
> |Factorized-FL|182.50|292.00|200.75|310.25|127.75|182.50|146.00|219.00|18.25GB|
> |MAPA$_{2k}$|0.32|-|0.94|-|0.32|0.79|0.56|-|**0.78MB**|
> |MAPA$_{16k}$|**0.08**|**0.18**|**0.23**|**0.45**|**0.08**|**0.18**|**0.12**|**0.27**|6.25MB|
> |MAPA$_{40k}$|3.84|8.64|10.88|21.12|3.84|8.64|5.76|13.12|0.32GB|
>
>
> ## 3. **Theorem 4.3**
>
> We apologize for the confusion regarding our convergence result. Our convergence bound matches FedAvg and recovers it as a special case: when reconstruction error is zero ($\epsilon = 0$), MAPA reduces exactly to FedAvg with the tightest convergence bound. For $\epsilon \neq 0$, the bound introduces a modest constant factor $(\epsilon + \beta + \epsilon\beta)$ due to compressed update distortion. Nevertheless, MAPA maintains the same asymptotic rate $\mathcal{O}(1/\sqrt{T})$ as FedAvg under standard assumptions (smoothness, bounded variance) [3,4]. Practically, this means MAPA might require slightly more rounds at higher compression, yet the total communication cost to achieve target accuracy significantly decreases, allowing training with substantially reduced overhead.
>
> ## 4. **References**
> [1] Jeong, W. and Hwang, S.J. "Factorized-FL: Personalized Federated Learning with Parameter Factorization & Similarity Matching."
> [2] Guo, P. et al. "Selective Aggregation for Low-Rank Adaptation in Federated Learning."
> [3] Yu, H. et al. "Parallel Restarted SGD with Faster Convergence and Less Communication."
> [4] Kim, D.-Y. et al. "Achieving Lossless Gradient Sparsification via Mapping to Alternative Space in Federated Learning."
> [5] Denil, M. et al. "Predicting Parameters in Deep Learning."
> [6] Li, C. et al. "Measuring the Intrinsic Dimension of Objective Landscapes."
> [7] Gressmann, F. et al. "Improving Neural Network Training in Low Dimensional Random Bases."
> [8] Aghajanyan, A. et al. "Intrinsic Dimensionality Explains the Effectiveness of Language Model Fine-Tuning."
> [9] Hameed, M.G.A. et al. "ROSA: Random Subspace Adaptation for Efficient Fine-Tuning."
> [10] Zhao, J. et al. "Galore: Memory-Efficient LLM Training by Gradient Low-Rank Projection."
> [11] Zhao, H. et al. "SEPARATE: A Simple Low-Rank Projection for Gradient Compression."
> [12] Kuang, W. et al. "Federatedscope-LLM: A Comprehensive Package for Fine-Tuning LLMs in Federated Learning."

---

> > ### Comment · Reviewer_Wq8k · 2025-04-07
> >
> > Thank you for your response and clarifications! I don’t have any further questions and will keep my score as it is.

---

### Official Review · Reviewer_NJUL · 2025-03-14

**Overall Recommendation:** 4

**Summary:**

This paper proposes Model-Agnostic Projection Adaptation (MAPA), which improves LoRA and FA-LoRA in federated learning (FL) by treating the entire model update as a single matrix rather than using layer-wise factorization. This approach enhances computational and communication efficiency while maintaining accuracy. MAPA introduces round-wise randomization of the reconstruction matrix to avoid suboptimal solutions and balance communication and accuracy. Unlike FA-LoRA, which uses a fixed A, MAPA regenerates A each round, enabling better parameter space exploration and preventing suboptimal convergence. Additionally, MAPA reduces memory and computational overhead compared to LoRA, ensuring greater efficiency in FL settings.

**Claims And Evidence:**

The main claim in this paper is:

- The proposed MAPA treats the entire model's weights as a single matrix and uses a unified low-rank space (Delta W and A) for low-rank
adaptation fine-tuning in FL.

The authors also claim that MAPA:

- Reduces communication costs compared to existing methods.
- Improves convergence through randomization of the reconstruction matrix.

The empirical experiments conducted on various benchmark datasets and tasks mostly support these claims.
However, I have two concerns regarding the evidence:

In some experimental settings, MAPA does not consistently outperform certain baselines in terms of convergence accuracy.
The paper lacks ablation studies to fully analyze the impact of the MAPA factorization process.

**Essential References Not Discussed:**

Not Available

**Experimental Designs Or Analyses:**

The proposed method and evaluation criteria are mostly appropriate, but the lack of ablation studies and deeper analysis of MAPA’s improvements. Additional experiments on more detailed ablation studies would strengthen the claims.

**Methods And Evaluation Criteria:**

This paper use popular benchmark models and datasets in FL for emperical experiments, which make sense.
My only 2 concerns are:
- Lack of ablation studies: While MAPA introduces randomized reconstruction matrices, the paper does not provide sufficient ablation experiments to isolate the impact of this randomization on convergence. A comparison between fixed vs. randomized reconstruction matrices would help clarify the exact benefits of the approach.
- FA-LoRA comparison: The comparison between FA-LoRA and MAPA should be extended and further elaborated. The authors could provide more context on FA-LoRA and explain why there is a performance gap between the two methods in certain settings.

**Other Comments Or Suggestions:**

1. Some experimental comparisons (e.g., with FA-LoRA) could be further elaborated
2. Conduct Ablation study to validate the proposed model agnostic factorization

**Other Strengths And Weaknesses:**

One notable strength of this paper is that the MAPA framework may not necessarily limited to FL, its model-agnostic factorization approach could be useful in regular centralized ML as well.

**Questions For Authors:**

The idea of unified low-rank space adaptation for fine-tuning is quite interesting. It seems like this approach could be useful not only in federated learning (FL) but also in traditional centralized fine-tuning. What makes this method particularly beneficial in FL settings?
Have similar ideas been explored in centralized ML?

**Relation To Broader Scientific Literature:**

Not sure

**Theoretical Claims:**

I checked the convergence proof, it make sense to me.

---

> ### Author Rebuttal · Authors · 2025-04-01
>
> ## 1. **MAPA does not consistently outperform certain baselines**
>
> Thank you for your careful observation. We want to emphasize that all the results provided in Table 2 and additional experiments during this rebuttal show that MAPA consistently outperforms in communication and performance.
>
> The results shown in Figure 5, in the top row, do not consider communication load. We are just comparing in terms of global rounds. As we take communication load into account, as shown in Table 2, we always perform better than baselines in performance per communication.
>
> &nbsp;
>
> ## 2. **The paper lacks ablation studies**
>
> Thank you for your constructive feedback. We initially provided our ablation studies on the effect of matrix rank on training (Figure 7) and the importance of fixed vs. fresh matrix A (Figure 6). Considering your comments, we additionally extended our studies on:
>
> ### 1. Fixed vs. fresh (randomization) of matrix A
> To elaborate on the effectiveness of randomization, additional experiments regarding MNIST and CIFAR10 are presented here, showcasing the accuracy across various ranks from $2^0$ to $2^{13}$, which clearly highlights the advantage of randomization, especially at lower ranks. Moreover, a discussion on the importance of randomization in training is located in **response 1 to reviewer Wq8k**.
> Method-Dataset/2^|0|1|2|3|4|5|6|7|8|9|10|11|12|13
> -|-|-|-|-|-|-|-|-|-|-|-|-|-|-|
> FrozenA-MNIST|7.93|9.43|9.83|16.86|19.36|42.09|69.94|81.57|92.85|95.17|96.46|96.91|97.84|97.86
> **FreshA-MNIST**|72.21|83.0|91.00|93.05|96.14|96.93|97.48|97.56|97.75|97.78|97.83|97.74|97.79|97.76
> FrozenA-CIFAR10|12.46|13.69|16.72|19.13|21.64|20.99|27.35|31.07|40.23|47.28|54.0|63.36|67.26|68.77
> **FreshA-CIFAR10**|51.53|55.02|57.95|61.37|63.82|65.5|66.5|69.2|68.62|69.02|68.31|68.34|68.71|68.59
>
> ### 2. Effect of rank in LLM fine-tuning
> We study the effect of MAPA rank on four different orders of magnitude alongside LoRA's baselines in communication-efficient LLM fine-tuning. The results are presented in the 2nd Table of **response 3 for reviewer FKCh**.
>
> ### 3. Experiments on layer-wise vs. model-level factorization
> Additionally, during the rebuttal, we conducted further experiments on LLM fine-tuning across various MAPA ranks, clarifying the trade-off between communication and performance, and additionally conducted an ablation study on layer-wise vs model-level factorization. (See **response 3 to reviewer JjTg**)
>
> If concerns remain, please specify any additional ablation studies you recommend. We remain committed to conducting further experiments.
>
> &nbsp;
>
> ## 3. **The comparison between FA-LoRA and MAPA**
>
> Methodologically, MAPA:
> 1. Factorizes gradients, not parameters (**response 1, reviewer FKCh**).
> 2. Uses a randomized $A$ instead of a fixed $A$, as shown in our ablations and **response 1 to reviewer Wq8k**.
> 3. Operates at the model level rather than layer by layer (**response 3, reviewer JjTg**).
>
> We further validated these claims via additional GLUE fine-tuning experiments against FA-LoRA and other baselines (**response 3, reviewer FKCh**).
>
> &nbsp;
>
> ## 4. **Centralized fine-tuning**
>
> Following your advice, we tested MAPA in a centralized setup and observed substantial gains over other baselines (**3rd table in response 3, reviewer FKCh**).
>
> &nbsp;
>
> ## 5. **Have similar ideas been explored in centralized ML?**
>
> The literature on low-rank gradient factorization in deep learning can start from:
> - [5] shows the inherent low-rank structure of gradients.
> - [6] examined intrinsic dimensionality by identifying the lowest-dimensional fixed random subspace enabling model convergence.
> Subsequent works [7–11] expanded on these concepts by training NN within randomly generated gradient subspaces.
>
> Although these approaches shows the efficacy of low-rank gradient factorization, they suffer from extensive memory overhead, as they represent the gradient as a single vector \(G^d\), where \(d\) is the number of model parameters, resulting considerable memory usage to construct the random transformation \(A^{d \times m}\).
>
> MAPA significantly differs from prior approaches by reshaping gradients before factorization. This simple yet effective modification achieves roughly \(k\)-fold reduction in computation and \(k^2\)-fold lower memory usage without compromising performance, supported by our theoretical and empirical analyses **(Appendix H and C.5)**. Additional discussion comparing gradient vs parameter factorization appears in **response 1 to Reviewer FKCh**.
>
> References are located in **response 4 to reviewer Wq8k**.
>
> &nbsp;
>
> ## 6. **What makes this method particularly beneficial in FL?**
>
> A primary challenge in FL is mitigating communication overhead. Our MAPA directly addresses this via low-rank gradient factorization integrated with efficient communication. While highly beneficial in FL, gradient reductions offer limited advantages in centralized settings, where gradient communication isn't required.

---

### Decision · Program_Chairs · 2025-05-01

**Decision:**

Reject

**Comment:**

This work studies communication efficient Federated Learning. It proposes a method that improves LoRA and FA-LoRA by treating the entire model update as a single matrix rather than using layer-wise factorization.

The final recommendations are quite mixed (4, 3, 3, 1). Reviewers recognize the importance of the problem and most reviewers recognize the soundness of techniques.

There are a few common questions raised by the reviewers.
Limited theoretical contribution (Wq8k, FKCh)
Novelty/ related works (Wq8k, FKCh, JjTg)
Larger datasets and more ablations (FKCh, JjTg)

We welcome that the authors added discussions, new datasets and new experiments in the rebuttal. But given the common concerns and expected sizable changes, we recommend the authors resubmit the work to a future venue with the improvements.

In addition, JjTg raised a question regarding why not layer-by-layer separation. After reviewing the rebuttal, the AC does not see that there is a fundamental problem that is not fixable. We suggest that the authors provide more discussions and justifications in the future version.